# Molecular mechanism of Activin receptor inhibition by DLK1

Daniel Antfolk [1], Qianqian Ming [1], Anna Manturova [1], Erich J. Goebel[2], Thomas B. Thompson[2] & Vincent C. Luca [1] ✉

Delta-like non-canonical Notch ligand 1 (DLK1) influences myogenesis, adipogenesis, and other aspects of human development through a process that is largely attributed to the downregulation of Notch signaling. Here, we show that DLK1 does not bind to Notch receptors or affect ligand-mediated Notch activation, but instead engages the TGF-β superfamily member Activin receptor type 2B (ACVR2B). The crystal structure of the DLK1-ACVR2B complex reveals that DLK1 mimics the binding mode of canonical TGF-β ligands to compete for access to ACVR2B. In functional assays, DLK1 antagonizes Myostatin-ACVR2B signaling to promote myoblast differentiation, rationalizing a mechanism for the role of DLK1 in muscle development and regeneration. Crosstalk between Notch and TGF-β is mediated by interactions between the transcriptional regulators SMAD2/3 and the Notch intracellular domain (NICD), and DLK1 inhibits SMAD2/3-NICD colocalization. These findings indicate that DLK1 acts directly on ACVR2B to inhibit signaling, whereas the observed effects on Notch may be an indirect result of DLK1 interference with NICD-SMAD complex formation.

The Notch pathway is a short-range signaling system that controls cell fate decisions in developing organisms. In mammals, the core Notch signaling components consist of four receptors (NOTCH1-4) and the activating ligands Delta-like 1 (DLL1), Delta-like 4 (DLL4), Jagged 1 (JAG1), and Jagged2 (JAG2). The mechanism of Notch activation by canonical DLL and JAG ligands has been thoroughly defined through structural, cellular, and genetic studies[1]. However, several non-canonical Notch ligands regulate development through poorly defined molecular mechanisms. For example, Delta-like non-canonical Notch Ligand 1 (DLK1), Delta/Notch-like EGF-related receptor (DNER), and DLL3 have each been suggested to modulate Notch signaling based on their structural similarity with canonical ligands[2], yet none of these proteins have been shown to form direct biochemical interactions with Notch proteins.

The non-canonical ligand DLK1 has been proposed to influence stem cell proliferation through the negative regulation of Notch signaling[3–5]. One popular model suggests that DLK1 acts as a decoy ligand that competes for NOTCH1 binding to DLL or JAG proteins[5]. This model is supported by indirect evidence, such as the downregulation of Notch target genes following DLK1 expression[5]. Interactions between DLK1 and NOTCH1 have also been described in two-hybrid assays[4,6,7], although the relevance of these studies is unclear since the reducing environment of the cytosol/nucleus does not allow for the formation of essential disulfide bonds in Notch receptors[8]. Furthermore, recent contradictory evidence has suggested that DLK1 may activate, rather than inhibit, Notch signaling to maintain populations of hematopoietic stem cells (HSCs)[9]. Besides Notch, DLK1 reportedly interacts with cysteine-rich fibroblast growth factor receptor (Cfr), insulin-like growth factor binding protein 1 (IGFBP1), Fibronectin, Activin receptor type 2B (ACVR2B), and over 40 other proteins from the BioPlex protein-protein interaction database[10–14]. Collectively, the above data highlight the lack of molecular-level clarity regarding DLK1 function.

During mammalian development, DLK1 regulates myogenesis, adipogenesis, and neurogenesis[15]. The DLK1 gene is expressed at high levels in various tissues during embryonic development and then

[1]Department of Immunology, Moffitt Cancer Center & Research Institute, Tampa, FL, USA. [2]Department of Molecular and Cellular Biosciences, University of Cincinnati, Cincinnati, OH, USA. ✉e-mail: vince.luca@moffitt.org

expression declines rapidly after birth[16]. DLK1 knockout mice exhibit skeletal malformations, increased adiposity, retarded growth with high perinatal mortality, and DLK1 mutations cause early onset puberty and obesity in humans[17–19]. DLK1 levels are crucial during development as mice with a higher dose of DLK1 suffer similar increased neonatal mortality as DLK1 knockdown mice. Mice expressing a triple dose of DLK1 result in severe developmental defects and embryonal lethality[20].

DLK1 is also an important regulator of muscle stem cell proliferation and differentiation[19]. Dysregulation of DLK1 causes muscle hypertrophy in callipyge sheep[21] and overexpression of DLK1 increases the muscle mass and diameter of muscle fibers in mice[22,23]. In myopathies and acute muscle injuries, the levels of DLK1 expression peak during muscle differentiation and fusion into myotubes. Despite the above data, the precise role of DLK1 in muscle tissue is currently unknown, with multiple phenotypes that cannot be explained by Notch signaling. In adult tissues, DLK1 expression is restricted to a few cell types, including regenerating muscle and certain stem and progenitor cells in the liver and pancreas[16]. DLK1 expression can be reactivated in response to injury or disease, and DLK1 is frequently overexpressed in cancer[24]. The broad role of DLK1 in development and disease, coupled with its restricted expression profile in adults, have implicated DLK1 as a potential therapeutic target in both cancer and muscle wasting diseases[5,25].

Here, we identify activin receptor type 2B (ACVR2B) as a direct physical binding partner for DLK1 from proteomics data. The crystal structure of the DLK1-ACVR2B complex revealed that DLK1 engages the canonical ligand binding site of ACVR2B, and we found that DLK1 influences myogenic differentiation by competing for ACVR2B binding with the canonical TGF-β ligand myostatin. By contrast, we determined that DLK1 does not bind to Notch receptors, nor did DLK1 influence ligand-mediated Notch activation in Notch signaling assays. Finally, we demonstrate that DLK1 can indirectly interfere with Notch/ACVR2B crosstalk, which may help reconcile conflicting data from previous studies linking DLK1 to Notch signaling.

## Results

### DLK1 does not bind Notch receptors or influence ligand-mediated Notch activation

To predict whether DLK1 forms canonical Notch-ligand interactions, we performed a conservation analysis comparing the extracellular domains (ECDs) of DLK1 and the Notch ligand JAG1. The DLK1 ECD contains six EGF-like repeats, and alignment of DLK1 and JAG1 revealed that DLK1 lacks the C2 and DSL domains required for JAG1-NOTCH1 interactions[26,27]. Furthermore, key Notch-binding residues in EGF1-3 of JAG1, including three bulky hydrophobic interface residues (Y255, H268 and W280 in JAG1), are substituted for various smaller amino acids (P47, S60 and G72) in DLK1 (Fig. 1A, and Supplementary Fig. 1A)[6]. Based on this analysis, we hypothesized that DLK1 either does not bind Notch or engages Notch using a different binding mode than DLL or JAG.

We used surface plasmon resonance (SPR) to directly test whether DLK1 interacts with NOTCH1. In this highly sensitive assay, we were unable to detect binding between the full-length ECDs of DLK1 and NOTCH1, even at concentrations greater than 10 micromolar (Fig. 1B, and Supplementary Fig. 1B, C). To determine whether DLK1 interacts with NOTCH1 on the cell surface, we compared the binding of recombinant soluble DLK1 (hereafter, soluble DLK1 or sDLK1 in figures) and the canonical ligand Delta-like 4 (DLL4) to NOTCH1-overexpressing U2OS cells using flow cytometry (Fig. 1C, and Supplementary Fig. 1D). We found that DLL4, but not DLK1, bound to NOTCH1-expressing cells. We were also unable to detect soluble DLK1 binding to U2OS cells expressing NOTCH2 and NOTCH3, indicating that DLK1 does not bind to any of the three most abundant Notch subtypes (Supplementary Fig. 1E).

We next evaluated the ability of DLK1 to influence Notch signaling using a fluorescent NOTCH1 reporter assay (CHO NOTCH1-Gal4 H2B-mCitrine system)[28]. Because surface-tethered DLL and JAG proteins can stimulate Notch activation, we tested whether DLK1 could similarly activate NOTCH1 when immobilized on tissue culture plates. We found that immobilized DLK1 did not affect signaling, whereas immobilized DLL4 ligands stimulated high levels of Notch reporter activity (Fig. 1D). To test whether the addition of soluble DLK1 affects signaling, we administered the soluble DLK1 to the reporter cells in the presence of immobilized DLL4. Soluble DLK1 did not inhibit DLL4-mediated Notch activation in this format, indicating that DLK1 does not positively or negatively affect ligand-mediated Notch signaling (Fig. 1D).

### DLK1 interacts with TGF-β superfamily receptor ACVR2B

Despite multiple studies implicating NOTCH in DLK1-mediated phenotypes, we were unable to detect functional interactions between DLK1 and NOTCH proteins[3,29,30]. To search for an alternative DLK1 receptor, we analyzed the Bioplex 3.0 interactome, a database of protein-protein interactions identified through affinity-purification mass spectrometry[14]. Out of 42 potential DLK1 binding partners, only ten were extracellular or transmembrane proteins, and only one, ACVR2B, is a known cell surface receptor[14] (Fig. 2A, and Supplementary Fig. 2A, B). ACVR2B is a transforming growth factor-beta (TGF-β) superfamily protein whose signaling influences a wide range of developmental processes[31,32], and it has previously been implicated as a target for DLK1 in myogenesis[13]. Given the overlapping functions of ACVR2B with Notch and DLK1 in multiple contexts, we hypothesized that ACVR2B has a high probability of forming direct biochemical interactions with the DLK1 protein[19,25,33–36].

We used confocal microscopy and SPR to experimentally assess DLK1-ACVR2B binding. We determined that recombinant ACVR2B-Fc binds to the DLK1-overexpressing U2OS cells but not to untransduced cells (Fig. 2B), and that recombinant ACVR2B-Fc and DLK1 bind with a steady state dissociation constant ($K_D$) of 1.4 μM (Fig. 2C). To cross-validate these observations, we generated a U2OS cell line overexpressing GFP$^{spark}$-tagged ACVR2B and confirmed that soluble DLK1 co-localized specifically with ACVR2B on the cell surface (Fig. 2D). In cells lacking ACVR2B expression as determined by lack of GFP-signal, we saw no staining with soluble DLK1 (Supplementary Fig. 2C). Canonical TGF-β ligands are dimeric and signal by inducing the formation of a 2:2:2 complex between the ligand and Type 1 and Type 2 receptors[37,38]. To assess whether DLK1 binds to Type 1 receptors, we also tested the binding of DLK1 to Activin Receptor Type 1B (ACVR1B). DLK1 did not bind to ACVR1B (Supplementary Fig. 2D), nor did it bind to the ACVR2B paralog ACVR2A (SFig. 2E). We further investigated DLK1 specificity in a fluorescence-based assay by expressing six type I receptors (ACVR1, ACVR1B, ACVRLK1, BMPR1A, BMPR1B and TGFBR1), five type II receptors (ACVR2A, ACVR2B, TGFBR2, BMPR2 and AMHR2) and one type III receptor (TGFBR3) on the surface of yeast (Supplementary Fig. 2F) and stained the cells with soluble DLK1 (Fig. 2E). Among this panel, DLK1 bound only to ACVR2B, indicating that DLK1-ACVR2B interactions are exceptionally selective compared to those of more promiscuous TGF-β ligands such as Activin[39] or BMP-2[40].

### Structure of the DLK1-ACVR2B complex

We mapped the ACVR2B-binding domains of DLK1 using biolayer interferometry (BLI). We determined that ACVR2B interacts with DLK1 EGF4-6, but not with EGF1-3 (Supplementary Fig. 3A), and SPR measurements further revealed that the EGF5-6 region binds to ACVR2B with comparable affinity (KD = 1.0 μM) to the full-length ECD (Supplementary Fig. 3B). We next determined the 2.7 Å resolution crystal structure of DLK1 EGF5-6 bound to ACVR2B to investigate how DLK1 modulates ACVR2B function (Fig. 3A, and Supplementary Fig. 3C, Supplementary Table 1). The structure revealed that EGF5 of DLK1

 

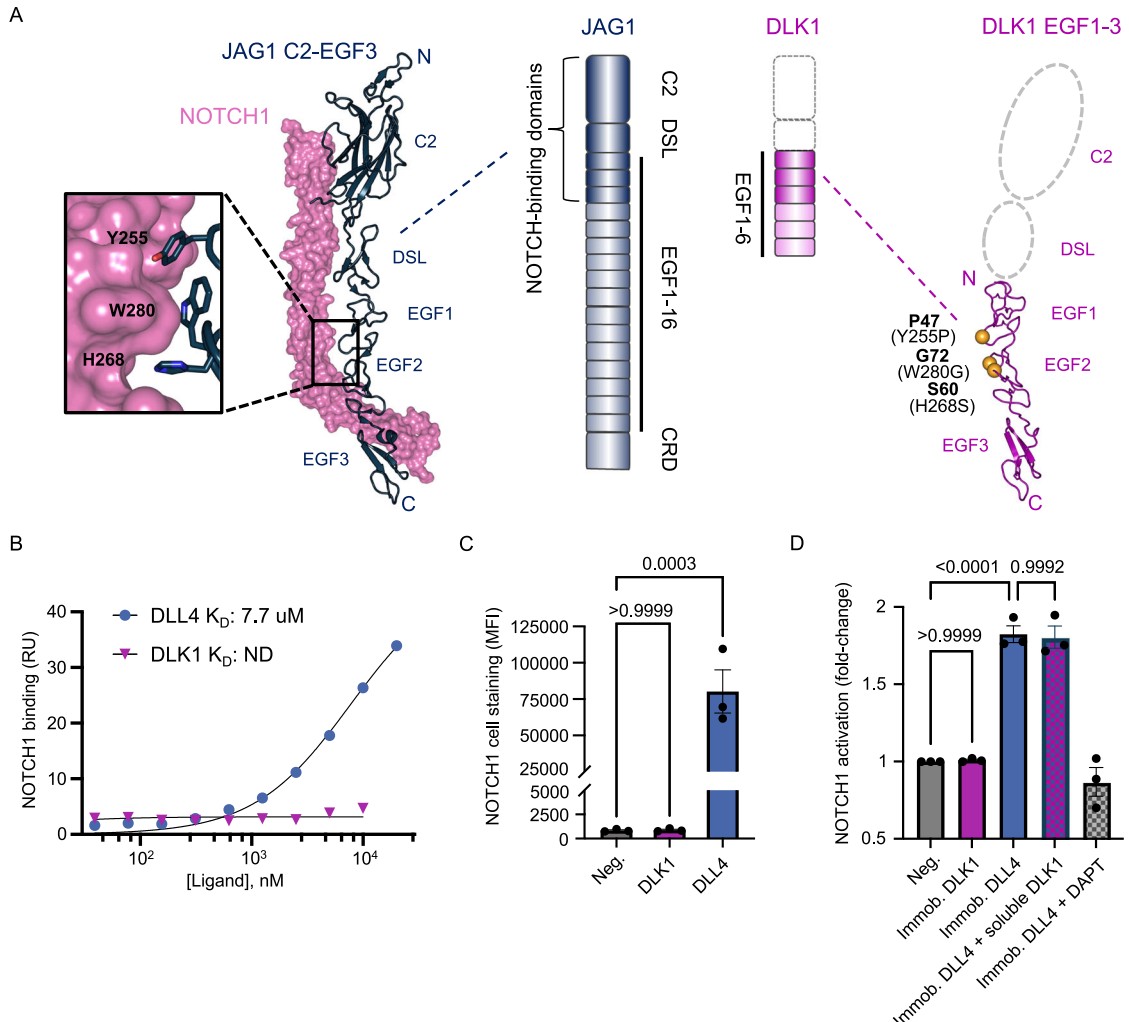

**Fig. 1 | DLK1 does not bind Notch1 or influence ligand-mediated Notch activation. A** Cartoon schematic showing the domain organization of the ECDs of JAG1 (dark blue) and DLK1 (magenta). The C2 and DSL domains that mediate interactions between Notch receptors and DLL or JAG ligands are absent in DLK1 (dotted lines). The structure of JAG1-NOTCH1 (left, PDB ID: 5UK5) has a zoom window showing Notch-binding residues that are not conserved in DLK1, and the analogous residues are depicted as yellow spheres in a structural homology model of DLK1 EGF1-3 (right). DSL = Delta/Serrate/LAG-2; EGF = Epidermal Growth Factor-like repeats; CRD = cysteine-rich domain. **B** SPR binding isotherm showing the binding of DLL4(N-EGF5) or the ECD of DLK1 to immobilized NOTCH1 (EGF1-36). The DLL4(N-EGF5) isotherm was fitted to a 1:1 binding model to determine the $K_D$. RU = resonance units. ND = not determined. **C** U2OS cells overexpressing NOTCH1 were stained with Fc-tagged DLK1 or DLL4 and binding was detected using an anti-Fc Alexa Fluor 647 antibody as measured by flow cytometry. Cells incubated with an anti-Fc-647 antibody alone was used as a negative control (Neg.). Bar graph depicts median fluorescence intensity (MFI) +/- SEM based on three independent biological replicates. Statistics were obtained using a one-way ANOVA in Prism 10

(Version 10.4.0) with Tukey's multiple comparisons post hoc test ($p > 0.9999$ (ns), 95% CI [−33686, 33589] for Neg. vs DLK1; $p = 0.0003$, 95% CI [−113045, −45769] for Neg. vs DLL4). **D** Notch activation and inhibition assay. Notch activation measured by flow cytometry using a CHO-K1 NOTCH1-Gal4 H2B-mCtirine reporter cell line with proteins immobilized on the bottom of the well (DLK1 N-EGF6, DLL4 N-EGF5) to activate plated NOTCH1 reporter cells. Soluble DLK1 was also used together with immobilized DLL4 activation to assess the ability of 3 µM soluble DLK1 to inhibit canonical Notch signaling. The γ-secretase inhibitor DAPT used at 3 µM was used as a Notch inhibition control. Bar graph depicts fold-change activation based on three independent biological replicates, where the control (Neg.) is depicted as 1 in each replicate. The control (Neg.) represents reporter cells alone. Statistics were obtained using a one-way ANOVA in Prism 10 (Version 10.4.0) with Tukey's multiple comparisons post hoc test ($p > 0.9999$ (ns), 95% CI [−0.2785, 0.2612] for Neg. vs Immobilized DLK1; $p < 0.0001$, 95% CI [−1.094, −0.5540] for Neg. vs Immobilized DLL4; and $p = 0.9992$ (ns), 95% CI [−0.2507, 0.2890] for Immobilized DLL4 vs Immobilized DLL4 + soluble DLK1. Source data are provided as a Source Data file.

forms the major ACVR2B-binding interface in the complex (Fig. 3A). Superimposing published structures of ACVR2B bound to Activin-A[41], BMP-2[42], or GDF11[43], as well as a structural homology model of Myostatin[44] bound to ACVR2B indicates that DLK1 occupies the binding site used by canonical ACVR2B ligands (Supplementary Fig. 3D, and Fig. 3B). Residues in the EGF5-EGF6 linker and EGF6 β-hairpin loop of DLK1 also make minor contacts with ACVR2B adjacent to the ligand-binding site. Canonical TGF-β ligands engage ACVR2B through the β-hairpin "fingers" of their cysteine knot domains (Supplementary Fig. 3D), and DLK1 appears to mimic the binding mode using the major β-sheet of EGF5 (Fig. 3A, and Supplementary Fig. 3D), with a slightly

larger binding interface (783.5 Å²) than canonical ACVR2B ligands (649.6 to 775.2 Å²) (Supplementary Table 2).

The DLK1 EGF5-ACVR2B interface is centered on W78 and F101 of ACVR2B, with the aromatic groups of these residues protruding into a pocket formed by the side chains of T186, I188, R193 and R195 of DLK1 (Fig. 3C, D, Supplementary Fig. 3E). Notably, W78 and F101 are part of a hydrophobic triad (Y60/W78/F101) that is critical for ligand-mediated signaling, and this region is completely occluded by DLK1[45]. We mutated several interface residues to determine their contributions to DLK1-ACVR2B interactions. We determined that the alanine substitutions W78A or F101A in ACVR2B were sufficient to ablate DLK1 binding

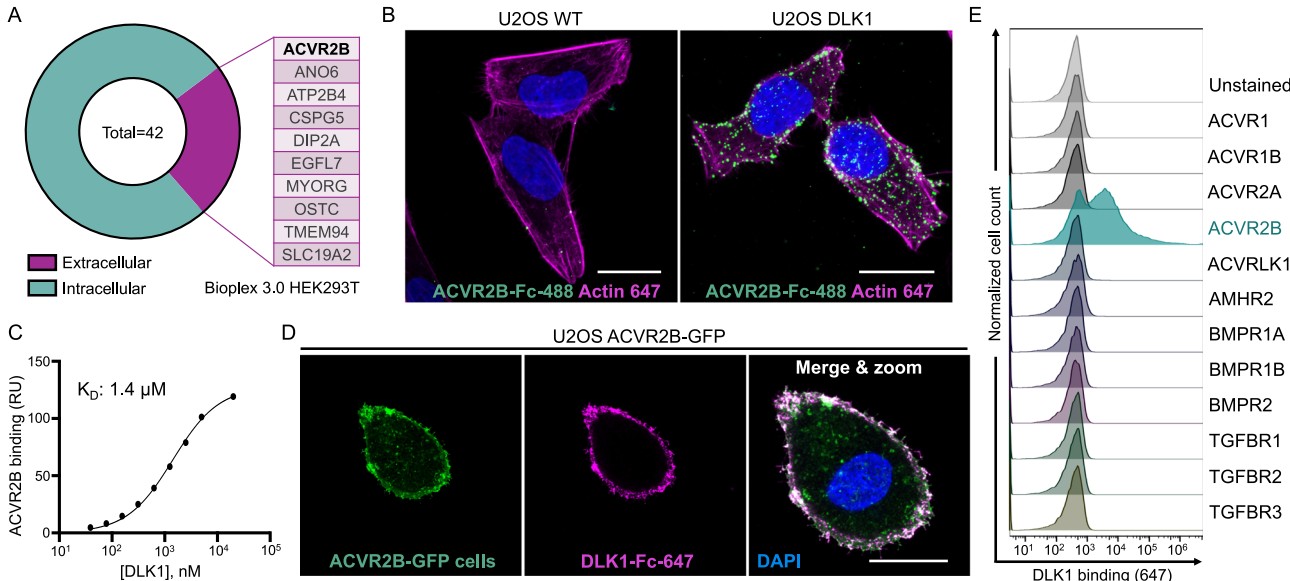

**Fig. 2 | DLK1 binds the TGF-β superfamily receptor ACVR2B. A** Analysis of the Bioplex 3.0 interactome[15] identified 42 potential DLK1 binding partners. The pie chart shows the fractions of candidate proteins containing extracellular domains (magenta) and intracellular proteins (teal). ACVR2B (bold) is the only known cell surface receptor among extracellular domain-containing proteins. **B** Confocal microscopy images depicting wild type U2OS cells or DLK1-expressing U2OS cells stained with recombinant ACVR2B-Fc protein. ACVR2B-Fc binding was detected with an anti-Fc Alexa Fluor 488 antibody, the contours of the cells were visualized by actin staining (magenta) using phalloidin 647, and nuclei were counterstained using DAPI (blue). The images are represented as maximum projections of 5 z-slices taken 0.8 μm apart. Scale, 20 μm. The experiment was independently repeated three times. **C** SPR was used to determine the steady-state binding affinity between DLK1(N-EGF6) and ACVR2B-Fc. The DLK1 protein was injected over a sensor chip containing immobilized ACVR2B-Fc and the data was fitted to a 1:1 binding model. RU = resonance units. The associated SPR sensograms for this data are shown in Supplementary Fig. 4A. **D** Confocal microscopy images depicting the staining of U2OS cells overexpressing ACVR2B coupled to a GFP$^{SPARK}$ (green) tag, with DLK1-Fc protein (magenta). DLK1 binding was detected using an anti-Fc Alexa Fluor 647 antibody. Nuclei counterstained with DAPI (blue). Scale, 20 μm. The experiment was independently repeated two times. **E** Flow cytometry histograms depicting the binding of DLK1-Fc protein to yeast expressing ACVR2B and eleven other TGF-β superfamily receptors. The experiment was independently repeated two times. Source data are provided as a Source Data file.

(Fig. 3E, Supplementary Fig. 4A-C). On the DLK1 side, a charge reversal mutation in EGF5, R193D, ablated DLK1 binding to both cellular and recombinant ACVR2B (Fig. 3F, Supplementary Fig. 4D-G) within cell-based co-localization assays and SPR, respectively.

Despite a high degree of amino acid conservation (73.6%) in the ECDs of ACVR2A and ACVR2B, DLK1 was unable to bind ACVR2A. To address this, we compared the structures of ACVR2A and ACVR2B to establish the molecular basis for preferential DLK1-ACVR2B interactions (Fig. 4A). In ACVR2B, F82 inserts into a groove formed by V209 and V229 of DLK1 (Figs. 3D, 4B). This F82 residue is substituted for a smaller isoleucine (I83) residue in ACVR2A, likely disrupting hydrophobic packing. The beta-branched isoleucine also clashes with a tightly packed loop around A198 in DLK1 (Fig. 4B). Additionally, T93 and E94 of ACVR2B are positioned opposite a positively charged region of DLK1, and these residues are substituted for lysines (K94, K95) in ACVR2A (Fig. 4C). These substitutions, in particular the charge reversal at E94 of ACVR2B (K95 in ACVR2A), would be predicted to perturb binding by altering the electrostatic character of the interface (Fig. 4D–G). Structural analysis of Myostatin (GDF8), GDF11, Activins A, B, C and E reveals that the charge reversal may also lead to biased receptor recognition by canonical TGF-β ligands (Supplementary Fig. 4H–I). This charge complementarity is unlikely to affect binding to the uncharged residues in Activin A/B, but it potentially explains preferential interactions between ACVR2B and GDF8/11, and between ACVR2A and Activin C/E (Supplementary Fig. 4H–I)[43,46,47].

## DLK1 inhibits ligand-mediated activation of ACVR2B
The inability of DLK1 to bind to Type 1 receptors, coupled with its steric occlusion of the ACVR2B ligand-binding site (Figs. 2E, 3B, and Supplementary Figs. 2D, 3D), led us to hypothesize that DLK1 functions

as an ACVR2B antagonist. ACVR2B signaling is important for muscle development and differentiation, and the TGF-β family ligand Myostatin negatively regulates myogenesis by activating ACVR2B[46,48–50]. Downstream of ACVR2B, the transcription factors SMAD2 and SMAD3 regulate this process by modulating the expression of genes such as MYOD, Myf5 and myogenin[51]. To test the effect of DLK1 on myostatin-ACVR2B signaling, we used a previously established HEK293 (CAGA)$_{12}$-luciferase reporter cell line to measure SMAD3-dependent activation through TGF-β responsive CAGA box repeats[52]. The clonally selected CAGA$_{12}$ reporter cells are highly sensitive to myostatin, which induces a 30 to 100-fold increase in luciferase signal compared to untreated cells (Supplementary Fig. 5A)[52]. Pre-incubating the cells with soluble DLK1 prior to myostatin treatment resulted in a dose-dependent inhibition of CAGA$_{12}$ luciferase reporter activity, with the highest tested concentration of DLK1 (16 μM) leading to ~ 95% inhibition (Fig. 5B, and Supplementary Fig. 5A). We also used SPR to test the ability of soluble DLK1 to interfere with myostatin-ACVR2B interactions and saw a similar level of dose-dependent inhibition (Supplementary Fig. 5B, C). Furthermore, we observed an inhibition of up to 50% upon DLK1 transfection in the same reporter cell line (Fig. 5C, and Supplementary Fig. 5D). This suggests that surface-expressed DLK1 is physiologically potent by achieving a local concentration at the surface comparable to a high concentration of soluble DLK1.

## DLK1 rescues myostatin-mediated blockade of muscle differentiation
We utilized a C2C12 myoblast differentiation assay to investigate whether DLK1 inhibits ACVR2B in a more functional setting. C2C12 cells differentiate into multinucleated myotubes when cultured in low serum media, and myostatin signaling through ACVR2B has previously

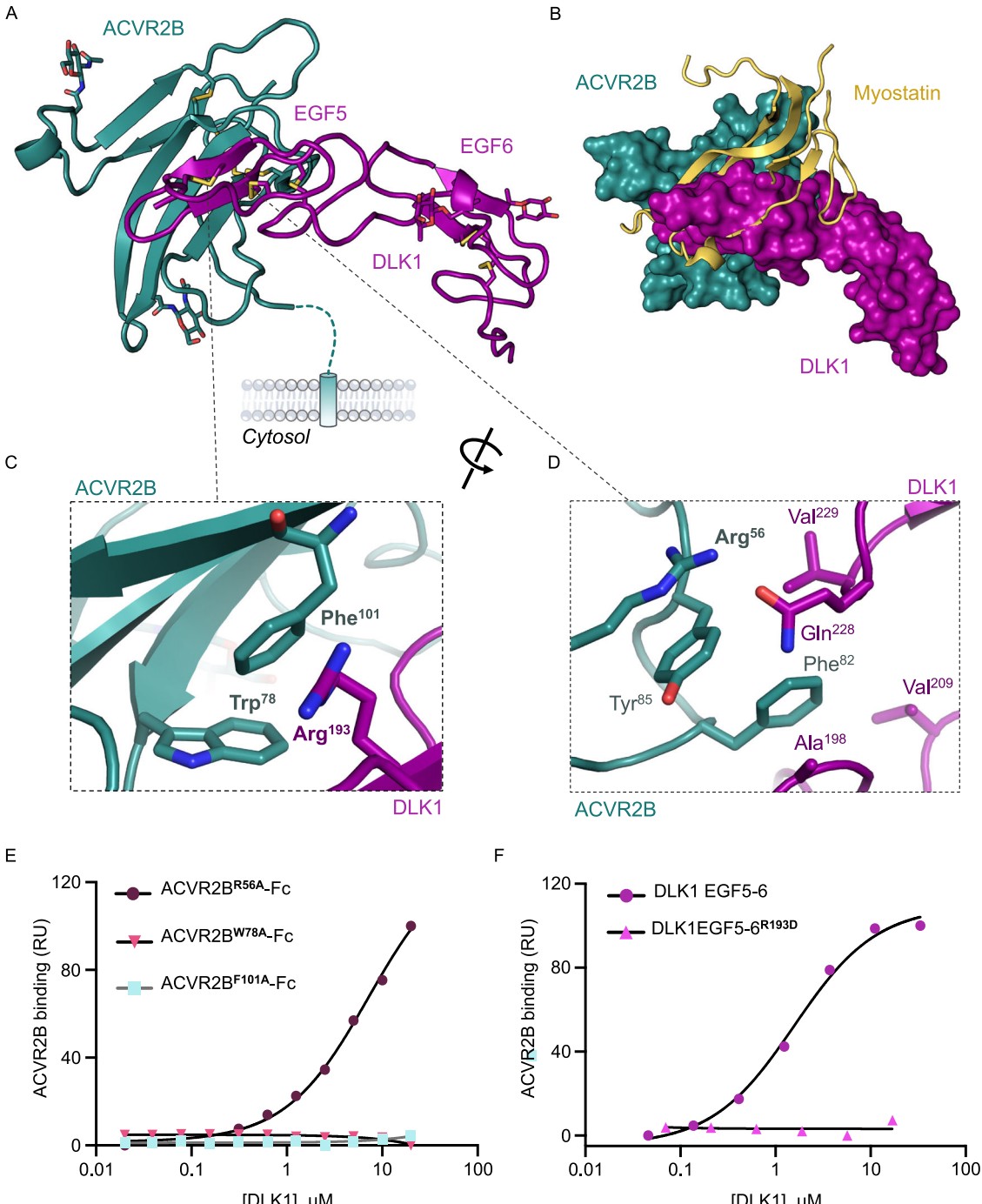

**Fig. 3 | Crystal structure of DLK1 EGF5-6 bound to ACVR2B. A** Crystal structure of DLK1 domains EGF5-6 (magenta) in complex with the extracellular domain of ACVR2B (teal). **B** A surface model of ACVR2B (teal) with DLK1(magenta) overlaid with a cartoon representation of myostatin (yellow). **C** Zoom in panel showing Trp[78] and Phe[101] of ACVR2B forming hydrophobic interactions with Arg[193] of DLK1. **D** Zoom in panel showing Phe[82] of ACVR2B packing against Val[209] and Val[229] of DLK1, and Arg[56] of ACVR2B forming a hydrogen bond with Gln[228]. **E** SPR isotherms comparing the binding between DLK1(EGF5-6) or DLK1(EGF5-6)[R193D]-mutant to ACVR2B-

Fc. The DLK1 proteins were injected over a sensor chip containing immobilized ACVR2B-Fc and the data was fitted to a 1:1 binding model. RU = resonance units. **F** SPR isotherms comparing the binding between DLK1 and three ACVR2B-Fc interface mutants. The R56A mutation in ACVR2B was associated with a ~ 6-fold decrease in DLK1-binding affinity compared to WT ACVR2B, and there was a complete loss of DLK1 binding to the W78A or F101A mutants. RU = resonance units. Source data are provided as a Source Data file.

been shown to inhibit this process[53]. C2C12 myoblast cells were cultured in differentiation media and analyzed for their ability to undergo myogenic differentiation. After 3 and 4 days of incubation, untreated C2C12 cells showed extensive formation of multinucleated myotubes and were positive for myosin heavy chain (MyoHC) expression (Fig. 5D, and Supplementary Fig. 6A, B). By contrast, C2C12 cells treated with

myostatin showed reduced myotube formation and MyoHC expression (Fig. 5D, and Supplementary Fig. 6B). In this assay, a DLK1 concentration of 2 μM was able to reverse the majority of the anti-differentiation effects of myostatin, which agrees with the concentration range in our reporter assay (Fig. 5B). Consistent with the lack of binding to ACVR2B, the DLK[R193D] mutant was unable to rescue the

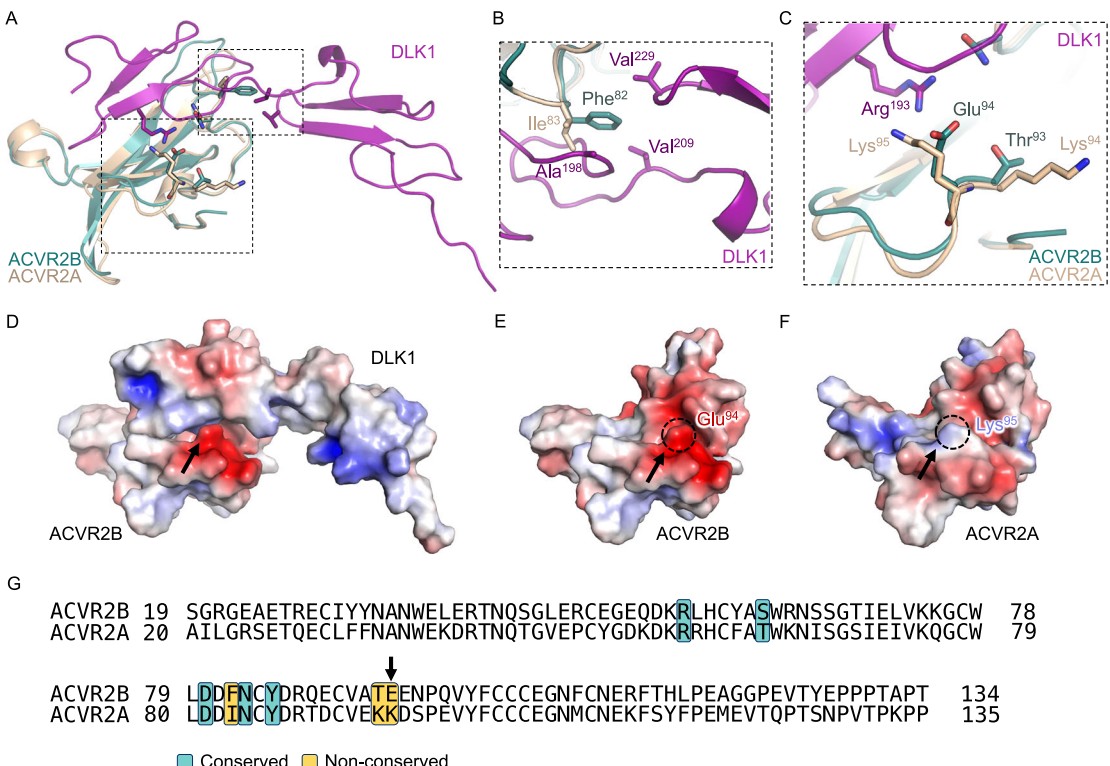

**Fig. 4 | Structural basis for highly selective interactions between DLK1 and ACVR2B. A** The structure of ACVR2A (PDB ID: 5NH3) was superimposed onto ACVR2B in the DLK1-ACVR2B complex structure (PDB ID: 9D20). **B** A zoom window shows Phe[82] of ACVR2B inserted into the pocket formed by DLK1 residues Val[209] and Val[229]. The analogous residue of ACVR2A, Ile[83], is not predicted to fit into this pocket. **C** A zoom window shows Arg[193] of DLK1 forming polar interactions with Thr[93] and Glu[94] of ACVR2B. The analogous residues in ACVR2A, Lys[94] and Lys[95], are not predicted to form charge complementary interactions. **D–F** Electrostatic potential surface representation of the DLK1-ACVR2B complex. The arrow indicates the ACVR2B interface glutamate (E94) residue that is substituted for a lysine (K95) in ACVR2A. **D** ACVR2B alone (**E**) and ACVR2A alone (**F**). **G** Sequence alignment of human ACVR2B and ACVR2A. Selected conserved (teal) and non-conserved (yellow) residues forming the DLK1-ACVR2B interface are highlighted. The E94 residue of ACVR2B is indicated with a black arrow.

inhibitory effects of myostatin treatment (Fig. 5D). Thus, DLK1 can modulate signaling from canonical TGF-β ligands through ACVR2B in multiple contexts. To compare the phenotype induced by soluble DLK1, we tested Bimagrumab, a high-affinity ACVR2B-blocking monoclonal antibody[54]. Bimagrumab similarly rescued myostatin-mediated disruption of myoblast differentiation (Supplementary Fig. 7A), but at significantly lower concentrations due to its high affinity. Combined treatment with soluble DLK1 and Bimagrumab showed no additive effect (Supplementary Fig. 7A), suggesting that both act through the same pathway.

### ACVR2B effector proteins SMAD2/3 interact with the intracellular domain of Notch

Although we showed that DLK1 does not directly interact with NOTCH, the perceived role of DLK1 in the Notch pathway may potentially be explained by crosstalk between intracellular Notch and TGF-β effector proteins[34,55,56]. It was previously shown that TGF-β signaling promotes interactions between activated SMAD3 and the Notch intracellular domain (NICD), and that TGF-β stimulates expression of Notch target genes through the direct binding of SMAD3 at two distinct sequences in the distal HEY-1 promoter region[57]. Therefore, we hypothesized that DLK1 could indirectly disrupt SMAD3/NICD association following myostatin stimulation.

We used proximity ligation assays (PLAs) to demonstrate physical interaction between SMAD2/3 and NOTCH1 in the C2C12 myoblast system (Fig. 6A). Distinct PLA punctae could be seen using NOTCH1 and SMAD2/3 antibodies with PLA probes (Fig. 6B), with no signal using either antibody alone with the PLA probes (Supplementary Fig. 8A–C). Furthermore, myostatin treatment for 2 h led to an 8-fold

increase in the number of PLA interactions observed between NOTCH1 and SMAD2/3 (Fig. 6B, C). Pre-treating the cells for 30 minutes with DLK1 before myostatin treatment resulted in a reduction of PLA signals similar to non-treated samples (Fig. 6B, C). Collectively, these PLA results corroborate previous data showing that NICD and SMAD interact, and that TGF-β family ligands can increase this interaction[55]. Our data suggests that, in addition to directly interfering with TGF-β ligand signaling through ACVR2B, DLK1:ACVR2B interactions could also indirectly regulate Notch target gene expression.

## Discussion

Our study indicates that descriptions of DLK1 as a non-canonical Notch ligand, both in published literature and gene nomenclature, do not fully reflect its biological function. This initial characterization may have biased the focus of DLK1 research towards Notch-related mechanisms, even in cases where Notch signaling cannot explain the results[10,58]. Therefore, we suggest that the role of DLK1 in development should be re-examined in the light of DLK1-mediated inhibition of ACVR2B.

From a molecular perspective, the DLK1-ACVR2B structure demonstrates how DLK1 is able to engage a Type II receptor through an EGF-like domain that is divergent from the cysteine-knot used by canonical ligands. This identifies DLK1 as a member of a small group of molecules that use alternative protein folds to modulate TGF-β superfamily signaling. For example, helminth parasites have been shown to secrete a sushi domain-containing protein, Hp-TGM, that activates TGF-β receptors to promote the expansion of regulatory T cells[59,60]. On the ligand side, the secreted BMP inhibitors Crossveinless 2 (CV-2) and Twisted Gastrulation (TWSG1) bind BMP proteins using a

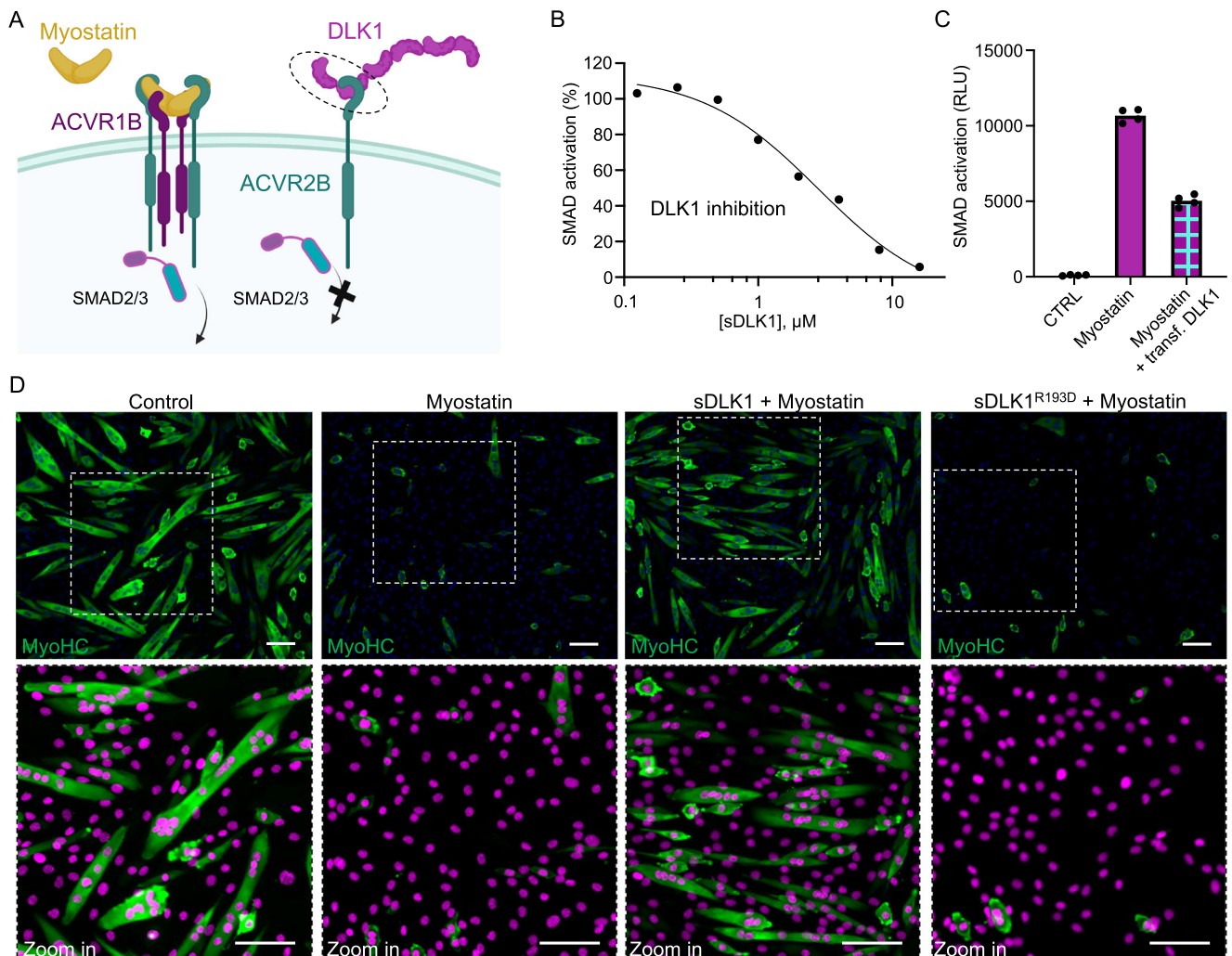

**Fig. 5 | DLK1 inhibits Myostatin-mediated ACVR2B activation in reporter cells and myoblasts. A** Illustration of Myostatin-ACVR2B signaling in the presence or absence of DLK1. Binding of the canonical ligand Myostatin to ACVR2B and ACVR1B leads to the formation of a 2:2:2 complex and subsequent activation of SMAD2/3 (left). The EGF5 domain of DLK1 binds to ACVR2B to inhibit ligand signaling (right), and the DLK1(EGF5-6) region used for co-crystallization is indicated with a dashed circle. Created in BioRender. Antfolk, D. (2025) https://BioRender.com/p73c456 **B** DLK1 inhibits Myostatin-ACVR2B signaling in a HEK293-(CAGA)$_{12}$ reporter assay. Myostatin treatment at 2 nM is represented as 100% activation, and a decrease in activation was observed upon treatment with increasing concentrations (125 nM-16 μM) of soluble DLK1. Data is represented as normalized relative luciferase units (RLU) represented as the mean of triplicate wells from one representative experiment. The experiment was independently repeated three times. **C** DLK1 transfected into HEK293-(CAGA)$_{12}$ reporter cells inhibit Myostatin signaling. Data is represented as RLU based on quadruplicate wells from one representative experiment. The experiment was independently repeated two times. **D** Representative microscopy images showing C2C12 myoblast differentiation in the presence of Myostatin, Myostatin + DLK1, or Myostatin + DLK1$^{R193D}$ (loss-of-ACVR2B-binding mutant). Control cells were allowed to differentiate for 72 h. Myostatin treatment (4 ug/ml) inhibits C2C12 myoblast differentiation into myotubes as determined by MyoHC staining. C2C12 cells were fixed with 4% PFA, immunostained with an anti-MyoHC antibody and an anti-mouse IgG Alexa Fluor 488 secondary antibody. Nuclei were counterstained with Hoechst 33342. Nuclei represented with pseudo color (magenta) in zoom in panels. Scale bar, 100 μm. The experiment was independently repeated four times. Source data are provided as a Source Data file.

Von Willebrand factor type C domain[61] and a helical cysteine-rich microdomain[62], respectively. Although we show that DLK1 and ACVR2B can interact in the absence of additional components, we cannot exclude the possibility of modulation by co-regulatory proteins or higher-order complex formation in a cellular environment. We also speculate that the intracellular domain of DLK1 may contribute additional regulatory functions. Future studies will be necessary to fully understand the full architecture and regulatory mechanisms of this complex.

We showed that DLK1 disrupts Myostatin-mediated activation of ACVR2B both in reporter cells and in a C2C12 myoblast differentiation assay. In a previous study, a soluble isoform of mouse DLK1 was an ineffective inhibitor of myoblast differentiation compared to a slight promoter by membrane-bound isoforms compared to untreated cells[63]. By contrast, our data show that DLK1 inhibits the negative effects of myostatin both in its cellular and soluble forms. In addition to the fundamental difference in experimental setup, we attribute this difference to the high concentrations of recombinant soluble DLK1 required for inhibition in our assays, as compared to the presumably lower concentrations secreted from transfected cells, although it's also possible that the multiple DLK1 isoforms found in mice have functions beyond the two isoforms found in humans. TGF-β ligands have been shown to bind ACVR2B with a wide range of affinities, with Myostatin and Activin A being among the most potent[45,64]. While canonical ligands have higher reported affinities for ACVR2B than DLK1[45], we showed that cellular DLK1 inhibits Myostatin-ACVR2B signaling, presumably due to its high local concentration in the membrane environment. Thus, we anticipate that DLK1 may function even more

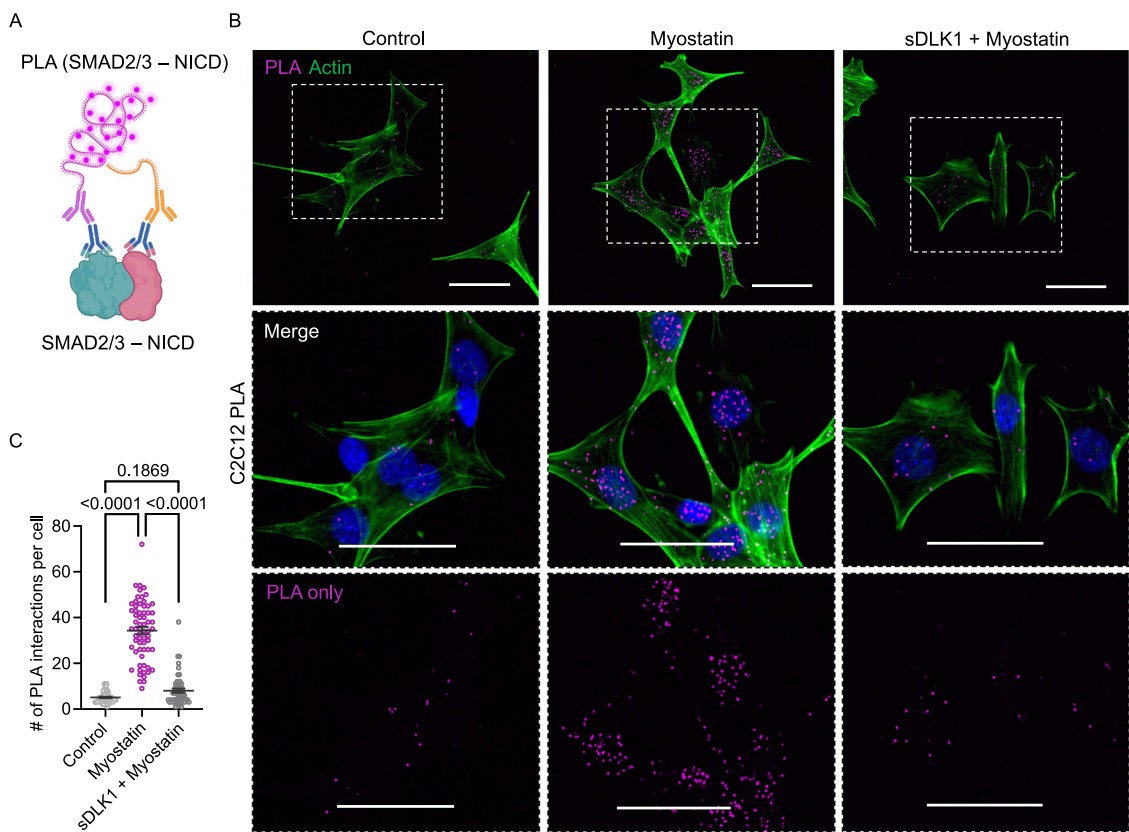

**Fig. 6 | The intracellular domain of NOTCH1 interacts with SMAD2/3 in C2C12 cells. A** Illustration of in situ proximity ligation assay (PLA), with antibodies targeting SMAD2/3 and NOTCH1. PLA uses secondary antibodies with oligonucleotides that can form a rolling circle amplification when both probes are in close proximity. Created in BioRender. Antfolk, D. (2025) https://BioRender.com/cyz9ujy **B** Immunofluorescence images showing the association between SMAD2/3 and NOTCH1 detected by in situ PLA. Untreated C2C12 cells (control), Myostatin-treated cells, and Myostatin + soluble DLK1 (sDLK1) treated cells were analyzed. Actin cytoskeleton stained by phalloidin 488. Nuclei counterstained by Hoechst 33342 (blue). Scale bar, 50 μm. The experiment was independently repeated three times. **C** Dot plot depicts quantification of manually counted PLA dots per cell +/− SEM from the experiments performed in Fig. 6B. The total number of cells counted per treatment, $n = 53$ (Control), $n = 65$ (Myostatin), and $n = 55$ (sDLK1 + Myostatin). Statistics were obtained using a one-way ANOVA in Prism 10 (Version 10.4.0) with Tukey's multiple comparisons post hoc test ($p < 0.0001$, 95% CI [−33.05, −25.35] for Control vs Myostatin; $p = 0.1869$ (ns), 95% CI [−6.985, 1.027] for Control vs soluble DLK1 + Myostatin; and $p < 0.0001$, 95% CI [22.41, 30.04] for Myostatin vs soluble DLK1 + Myostatin). Source data are provided as a Source Data file.

efficiently as a selectivity filter against lower-affinity ligands such as Activin C/E, BMP-2, BMP-3 and BMP-7[64]. This is supported by studies of DLK1 in neuronal differentiation, as DLK1 expression has been shown to promote neurogenesis by interfering with BMP/SMAD signaling[65]. Myostatin primarily signals through the canonical SMAD2/3/4 complex to regulate differentiation factors such as Atrogin-1, MyoD, and MYOG, and through non-canonical SMAD-dependent pathways involving ERK1/2 and other MAPK effectors[46,66,67]. These established myostatin mechanisms are likely antagonized by the DLK1-ACVR2B interaction. However, we also consider it plausible that a synergistic effect, through NICD-SMAD complexes, could further contribute to DLK1's regulatory effects of muscle differentiation[68].

In addition to myogenesis and neurogenesis, other systems are likely to be regulated by DLK1-mediated inhibition of ACVR2B signaling. In the adipogenesis field, DLK1 has mostly been shown to inhibit adipogenesis, although a subset of studies suggest that DLK1 may promote adipogenesis in certain contexts[15–17,69]. While the precise molecular mechanism of DLK1 in adipogenesis has not been determined, we note that the transcription factor PPARγ (peroxisome proliferator-activated receptor γ) is both essential for adipocyte differentiation[70] and downstream of TGF-β ligand signaling[71–73], suggesting that DLK1 may play a role in adipogenesis by antagonizing ACVR2B[74]. DLK1 may also function indirectly in adipogenesis by preventing ligand-induced assembly of ACVR2B and ACVR1C (also known

as ALK7), as ACVR1C signaling was recently shown to inhibit PPARγ expression and adipogenesis following stimulation with Activin-E[75,76]. We speculate that DLK1 may have differential effects depending on the presence of high- or low-affinity ACVR2B ligands based on its moderate affinity for ACVR2B. However, this will need to be tested experimentally in future studies.

It is notable that DLK1-ACVR2B interactions are highly selective, and that DLK1 does not bind to ACVR2A or other TGF-β family receptors. We posit that this selectivity enables DLK1 to facilitate tissue-specific regulation. Of the two receptors, ACVR2A is predominantly expressed in osteoblasts[77]. Mice lacking ACVR2A show increased femoral trabecular bone volume due to increased osteoblast differentiation, whereas ACVR2B knockdown mice have no significant changes in bone parameters[77]. Conversely, ACVR2B knockdown in chickens leads to significantly higher body weight compared to knockdown of ACVR2A alone or in combination with myostatin, indicating ACVR2B can have an outsized effect on muscle tissue[78]. This latter effect is consistent with recent data indicating that antibody-mediated blockade of ACVR2A/B signaling preserves muscle mass during the use of popular GLP-1 receptor agonist weight loss drugs[54]. As DLK1 functions as a selective inhibitor of ACVR2B, it may have the potential to be adapted as a more targeted therapy for counteracting the loss of muscle associated with anti-obesity drugs, muscle wasting disorders, or cachexia.

## Methods

### Protein expression and purification

Human DLK1(ECD) (amino acids 24-260), DLK1(EGF1-3) (amino acids 24-127), DLK1(EGF4-6) (amino acids 128-260), and DLK1(EGF5-6) (amino acids 169-260), were cloned into pAcGp67A with a 3C protease cleavable C-terminal biotin-acceptor peptide tag (BAP-tag: GLNDIFEAQKIEW) and 6xHis tags. Human DLL4 (N-EGF5, amino acids 27-400) and Notch1(EGF1-36) (amino acids 20-1426) were each cloned into the pAcGp67A vector with a C-terminal 8xHis tag. DLK1 Fc-fusion constructs were obtained by cloning DLK1 ECD or EGF5-6 to pAcGp67A vector between a N-terminal human Fc-tag and a C-terminal His-tag. ACVR2B ECD (amino acids 24-136) and mutants were cloned to pAcGp67A followed by a 3C protease cleavage site and a human IgG Fc tag (ACVR2B-Fc). ACVR2B mutants were generated using site-directed mutagenesis and were cloned to the same vector as ACVR2B-Fc. All proteins were expressed using baculovirus by infecting 1 L of Tni insect cells (Expression Systems, Cat# 94-002 F) at a density of $2 \times 10^6$ cells/mL. Cultures were harvested after 60 h of inoculation. Proteins were purified from supernatants using nickel chromatography. Protein enriched on Nickel-NTA (Qiagen) was washed with W buffer (HBS: 20 mM HEPES pH 7.4, 150 mM NaCl and 10 mM imidazole), and was eluted with E buffer (HBS plus 250 mM imidazole). Notch1 proteins were supplemented with 1 mM calcium in the wash and purification buffers. The concentrated protein sample was subsequently applied on size-exclusion chromatography. BAP-tagged proteins and ligands used in binding experiments were site-specifically biotinylated at the C-terminal BAP tag with BirA ligase. NOTCH1(EGF1-36) protein was chemically biotinylated at the N-terminus using the EZ-Link™ Sulfo-NHS-Biotin reagent (ThermoFisher).

### Protein proteolytic processing and deglycosylation of ACVR2B

For crystallization, the DLK1 EGF5-6 protein was treated with 3C protease to remove the C-termini Tags followed by SEC in HBS buffer to remove residual 3C and cleaved tags. The ACVR2B-Fc protein was processed enzymatically to remove the C-terminal Fc tag and N-linked glycans. Degylcosylation of ACVR2B was adapted from a previously published protocol[79]. Briefly the ACVR2B protein intended for deglycosylation was expressed from cells cultures supplemented with 5 μM kifunensine at infection. Kifunensine-sensitized ACVR2B were purified and treated with 3C protease as described above. The tag-free ACVR2B was then incubated at 4 °C overnight with 1:100(w/w) Endoglycosidase F1 for removal of N-linked glycans. The processed ACVR2B was then mixed with DLK1 at 1:1 ratio. The mixture of proteins was treated with 1:100(w/w) bovine carboxypeptidase A and B (Sigma) at 4 °C overnight before they were applied to SEC to separate the complex fraction.

### Crystallization of DLK1-ACVR2B complex

The final SEC purified complexes were concentrated to ~18 mg/mL in HBS and crystallized by sitting drop vapor diffusion. DLK1(EGF5-6)-ACVR2B crystals grew from drops containing 0.1 μL of protein combined with 0.1 μL of mother liquid consisting of 2.1 M AmSO4, 0.1 M HEPES PH 7.6 and 2.5% polyethylene glycol (PEG) 400. The crystals were transferred into a cryoprotectant drop containing reservoir supplemented with 25% Ethylene Glycol. The dataset was collected from crystals grown in 2.1 M ammonium sulfate, 2.5% PEG 400, 0.1 M HEPES, pH 7.6.

### Data collection and structure determination

The dataset used for structure determination was collected at Advanced Photon Source beamline 22-ID. Data were indexed, integrated, scaled and merged using XDS. The phasing was solved using molecular replacement (MR) using the published structure of ACVR2B (PDB ID: 6MAC) as an initial search model. Individual models for DLK1 EGF5 and EGF6 were built from the published structure of the DLL1 EGF5 and EGF6 (PDB ID: 4XBM), respectively. The model prediction and building were performed using SWISS-MODEL server[80–84]. The MR was performed in PHENIX using Phaser[85,86]. Phaser model searching correctly placed ACVR2B and EGF5, and then EGF6 was manually built in the electron density map. Manual building was performed in Coot. Phenix.refine was used for refinement[87–89]. TLS parameters were applied at the end of refinement progress[90,91]. The final refinement gave a $R_{work}$ of 22.3% and $R_{free}$ of 24.7%. The structure contains two copies of DLK1:ACVR2B complex, with the better resolved copy representing DLK1 aa 173-251 and ACVR2B aa 26-118, while the second copy included DLK1 sequence 173-248 and ACVR2B sequence 26-118.

### Mammalian cell culture

C2C12 cells were from American Type Culture Collection (ATCC; Manassas, VA) (Cat# CRL-1772). HEK293T cells were a gift from Dr. Eric Lau (Moffitt Cancer Center, FL, original commercial source ATCC, Cat# CRL-3216). Notch reporter cell lines CHO-K1 N1-Gal4 were a gift from Dr. M. Elowitz (California Institute of Technology, original commercial source Invitrogen, Cat# R71807). U2OS and U2OS NOTCH1 reporter cells[92] were a gift from Dr. Stephen Blacklow (original commercial source ATCC, Cat# HTB-96). Human embryonic kidney 293 (HEK293) cells stably overexpressing DLK1, U2OS stably overexpressing DLK1, ACVR2B or GFPSpark control were grown in DMEM (Gibco), supplemented with 10% FBS, penicillin (100 units/mL), and streptomycin (100 μg/mL). U2OS DLK1, ACVR2B and GFPSpark cell lines were kept under selection with 200 μg/ml hygromycin B. C2C12 were purchased from ATCC (CRL-1772) and were cultured in DMEM (Gibco) (without sodium pyruvate) supplemented with 10% FBS, 2 mM glutamine, penicillin (100 units/mL), and streptomycin (100 μg/mL). For myoblast differentiation assays, 'differentiation media' consisting of DMEM + 2% horse serum (Gibco) was used to induce differentiation of C2C12 myoblasts for 72-96 h.

Generation of ACVR2B stable cell line. ACVR2B pCMV3-C-GFPSpark or pCMV3-C-GFPSpark control vector (SinoBiological, Cat# HG10229-ACG and Cat# CV026) were transfected using Lipofectamine 3000 in U2OS cells and stable cell lines were selected using 200 μg/ml hygromycin. Stable cell lines were further sorted based on GFPSpark-expression using a Sony SH800S cell sorter to generate ACVR2B and GFPSpark control cells with equal levels of GFP-fluorescence.

Generation of DLK1 stable cell line. cDNAs encoding human DLK1 was cloned into the pLenti-C-Myc-DDK (ORIGENE) lentiviral vector and sequence verified. U2OS cells were transfected using Lipofectamine 3000 and selected using 2ug/ml puromycin. The cells stably expressing DLK1 (U2OS DLK1) were sorted on FACS with anti-DLK1 Alexa Fluor 488 antibody at 1:1000 (R&D systems, FAB1144G-025). Cells were gated to collect the FITC-positive population.

HEK293-CAGA cells previously generated and published by Dr. Thomas Thompson[52]. A plasmid containing a PGK-neomycin cassette inserted into the pGL3-(CAGA)12-luciferase reporter construct[93] in the same orientation as the promoter using SalI/XhoI was generously provided by Dr. Alexandra McPherron. This construct was digested with SalI, gel-purified, and transfected into HEK293 cells that were ~50% confluent in a 6-well plate using TransIT-LT1 transfection reagent (Mirus). Clonal selection was carried out in DMEM plus Pen-Strep, 10% FBS, and 100 μg/ml G418. A stable cell line (HEK293 (CAGA)12) was derived by selecting the best clone based on the highest response to 0.8 ng of activin A in a 96-well plate.

### Cell based binding assays using flow cytometry

U2OS or U2OS DLK1 cells were resuspended in blocking buffer (1% BSA in PBS buffer pH 7.4, Gibco). Recombinant ACVR2B-Fc was preincubated with anti-Fc 647 at 1:120 (Goat Anti-Human IgG Fc-AF647, Southern Biotech, Cat# 2048-31) for 1 h under rotation at 4 °C. $0.5 - 1.0 \times 10^5$ cells were then incubated with 50–100 μl ACVR2B-Fc-647 (100 nM) for 1 hour at 4 °C. After washing the cells with blocking buffer followed by a 2 min centrifugation at 400 g and resuspending

the pellet three times, the cells were acquired with an Accuri™ C6 (BD, Bioscience) flow cytometer. The data was analyzed using FlowJo® v10.6.0 (BD Biosciences). For U2OS cells, DLK1-Fc, DLK1$^{R193D}$-Fc or DLL4-Fc protein was pre-incubated with anti-Fc 647, or anti-Fc 647 alone as a negative control (Southern Biotech, Cat# 2048-31) for 1 h under rotation at 4 °C. For U2OS NOTCH1 expressing cells, the NOTCH1 expression was first induced for 24 h with 2 µg/ml doxycycline added to the growth media. $1.0 \times 10^5$ cells were then incubated with 100 µl Fc-647-labelled protein using 3 µg/ml protein and 1:120 secondary antibody for 1 h at 4 °C. The cells were washed and acquired with an Accuri™ C6 (BD Biosciences) flow cytometer as described above. The data was analyzed using FlowJo® v10.6.0 (BD Biosciences). The gating strategies for flow cytometry can be found in Supplementary Fig. 9A–D and Supplementary Fig. 10A–C.

### Immunofluorescent cell staining

Cells grown on coverslips were fixed in 3% paraformaldehyde and permeabilized with 0.15% Triton X-100 in PBS for 5 min at RT. Nonspecific binding was blocked by incubation in 3% BSA in PBS with 0.05% Triton X-100 and 0.1 M glycine for 60 min at RT. Cells were stained with primary antibodies overnight at 4 °C or 2 h at RT, after which coverslips were rinsed three times with PBS and stained for 60 min with fluorescent tag-labeled secondary antibodies (Invitrogen Alexa Fluor). For visualization of protein binding, proteins with Fc-tags were preincubated with anti-Fc 488 (Rat anti-human IgG Fc Alexa Fluor 488, BioLegend, Cat# 410705, 1:100) or anti-Fc 647 (Goat Anti-Human IgG Fc-AF647, Southern Biotech, Cat# 2048-31, 1:400) for 1 h on rotation in +4 °C. Cells were washed three times in PBS before mounting with VECTASHIELD® including DAPI (Vector laboratories). Glass slides were imaged with a Leica SP8 laser scanning confocal microscope (Leica Microsystems GmbH, Germany). Images were acquired through a HC PL APO CS2 63x/1.4 oil objective. Images were analyzed with LAS X software version 3.7.4 (Leica Microsystems GmbH, Germany). For U2OS DLK1 and U2OS WT cells stained with ACVR2B-Fc in Fig. 2B, cells were blocked with DMEM + 10% goat serum and 1% BSA for 1 h and the cells were counterstained for filamentous actin with Alexa 647 conjugated to phalloidin (Phalloidin Alexa Fluor 647, Invitrogen, Cat# A22287, 1:400). Z-stack maximum projections using 5 images taken 0.8 µm apart were generated and exported as TIF files with LAS X (Leica Microsystems). For differentiation and PLA assays with C2C12 cells, the cells were plated on 24 well black frame plates with glass-like polymer bottoms (Cellvis). Proximity ligation assays were performed by incubating cells for 30 min with 2 µM recombinant soluble DLK1 before adding 2 µg/ml (128 nM) recombinant active human myostatin (Abcam, Cat# ab269163 and Cat# ab256090) for 2 h before fixation of cells. Nonspecific binding was blocked by incubation in 3% BSA in PBS with 0.05% Triton X-100 and 0.1 M glycine for 60 min at RT and by the Invitrogen PLA block solution for 30 min at RT. The protocol for PLA was then followed according to the manufacturer's specifications (Invitrogen) using primary antibodies Notch1 A-8 (Santa Cruz Biotechnology, Cat# sc-376403) and SMAD2/3 (Cell Signaling, Cat# 8685) at 1:100 concentration. For C2C12 differentiation assays, cells were plated in 24 well plates (Cellvis) and at ~70% confluency the experiment was started (Day 0) by replacing growth media with DMEM + 2% horse serum (differentiation media) and blocked for 30 min with recombinant soluble DLK1, DLK1$^{R193D}$-mutant or ACVR2B blocking antibody (Bimagrumab, MedChemExpress, Cat# HY-P99355) (100 nM) before addition of 4 µg/ml recombinant active myostatin (R&D systems). After 48 h, media was replaced by fresh differentiation media with or without soluble DLK1 and with or without myostatin at the same concentrations as day 0. The assay was concluded on day 3 or day 4 depending on the final confluency and extent of myotube formation in the control cells. Cells were fixed, permeabilized and blocked in the same way as the immunofluorescent cell stainings before incubating with anti-MyoHC antibody (R&D systems, Cat# MAB4470) at 1:400.

Cells were then washed 3 times in PBS + 0.5% BSA before 1 h incubation with anti-mouse Alexa Fluor 488$^{Plus}$ at 1:1000 (Invitrogen, Cat# A32723). Hoechst 33342 (Invitrogen, Cat# H1399) was used to counterstain nuclei at 1:5000 for 10 min. Nuclei were pseudo colored to magenta in the Keyence BZ-X710LE analyzer software in selected zoom in panels to highlight multinucleated myotubes. C2C12 differentiation images were acquired using a Keyence BZ-X710 microscope using a Nikon Plan Apo 10x objective and PLA images with a Nikon S Plan Fluor 40x objective. PLA images were processed with the de-haze function for the far-red channel in the BZ-X710LE analyzer software. Quantification of PLA punctae were counted from at least 50 cells per group. Detection of nuclei by Hoechst 33342 at 1:5000 (Invitrogen, Cat# H1399) and filamentous actin cytoskeleton staining by Alexa Fluor Plus 488 Phalloidin 1:400 (Invitrogen, Cat# 12379) were used to segment individual cells within each image.

### Luciferase reporter assay

HEK293 (CAGA)$_{12}$ reporter cells were plated in tissue culture treated clear bottom 96-well white plates (Costar Ref3610, Corning Incorporated) with or without poly-D-lysine (Cultrex). $4 \times 10^4$ cells were plated on day 1 in growth media. The next day cells were washed once with PBS and serum starved overnight. On day 3, cells were treated with varying amounts of DLK1 and 2 nM myostatin in 100 µl serum free media for 20–24 h. Following treatment, assay plates were equilibrated to room temperature for 15 min after which cells were lysed with 100 µl/well of Bio-Glo™ luciferase assay reagent. Cells in Bio-Glo reagent were incubated on a shaker plate for 15 min at RT before measuring luminescence on a Promega GloMax® Discover plate reader. Data was visualized in GraphPad Prism 9 (GraphPad Software, LLC, Version 9.5.1). RLU values presented in bar graphs in Fig. 5C and Supplementary Fig. 5A, and normalized values with myostatin treatment set as 100% for the DLK1 inhibition dose titration curve in Fig. 5B.

### Yeast surface display

The extracellular domains of twelve TGF-β family receptors were individually cloned into a modified pCT vector as N-terminal fusions to a c-Myc epitope (EQKLISEEDL) and the yeast cell wall protein Aga2. Plasmids were transformed into *Saccharomyces cerevisiae* EBY100 cells by electroporation and cultured in SD-CAA medium at 30 °C. After overnight growth and one passage, expression was induced in SG-CAA medium at 20 °C for 48 h. Receptor expression was detected using an Alexa Fluor 488–conjugated anti-Myc-Tag (9B11) antibody at 1:100 dilution (Cell Signaling Technology, Cat# 2279). Recombinant soluble DLK1-Fc protein was pre-incubated with Alexa Fluor 647-conjugated secondary antibody (Southern Biotech, Cat# 2048-31) for 1 h at 4 °C. The fluorescence-labelled protein (100 nM protein, 1:100 secondary antibody) was then incubated with $10^6$ yeast cells in 100 µl PBS for 1 h at 4 °C. Fluorescence was analyzed using an Accuri™ C6 flow cytometer (BD Biosciences), and data were processed using FlowJo® v10.6.0 (BD Biosciences). Gene fragments used for yeast display and primer sequences added as separate Supplementary Data 1.

### SPR binding studies

Dissociation constants between the DLK1 ECD or DLK1 EGF5-6 and ACVR2B binding were determined by surface plasmon resonance using a BIAcore T100 instrument (GE Healthcare). ~400 resonance units (RU) of recombinant wild-type or mutant ACVR2B or ACVR2A with Fc-His-tags were immobilized on a CM5 sensor chip using amine coupling. Increasing concentrations of DLK1 ECD or DLK1 EGF5-6 were used as analytes in HBS supplemented with 0.005% surfactant P20 and 0.1% BSA (HBS-P20-BSA) at 20 °C. Binding and dissociation were performed at least 45 µl/min for 90 sec and 120 sec, respectively. Each sample-injection cycle was followed by a 30-sec injection of regeneration buffer (0.5 M MgCl$_2$). Curves were reference-subtracted from a flow cell immobilized with the negative control protein (T-cell

immunoglobulin mucin-3, Tim-3). The equilibrium RU was plotted as a function of concentration using Prism 9 (GraphPad). Steady-state binding curves were fitted using BIAcore T100 evaluation software to a 1:1 Langmuir model to determine the $K_D$. For measurement of binding between DLK1 and NOTCH1, ~300 RU of NOTCH1(EGF1-36) or the negative control protein (extracellular domain of human ZNRF3) were immobilized on a streptavidin coated sensor chip (GE Healthcare). Either recombinant DLL4(N-5) protein (positive control) or DLK1 ECD protein was used as analyte in HBS-P20-BSA supplemented with 1 mM $CaCl_2$. In the DLK1-Myostatin competition assay for ACVR2B binding, 2-fold serial dilutions of recombinant DLK1 protein (starting at 30 µM) were mixed with a constant concentration of 0.8 µM recombinant ACVR2B-Fc protein and used as analytes with immobilized GDF8 (myostatin) protein (R&D) on a CM5 chip. The data processing was the same as described for DLK1-ACVR binding.

### Biolayer interferometry (BLI) assays
BLI assays were performed using the Gator™ Label-Free Bioanalysis instrument (Gator Bio, Palo Alto, CA, USA) and Gator One 2.7 (software version 2.15.5.1221). Streptavidin (SA) sensor probes (Gator Bio, USA, Cat# #160002) were used for the sample measurements. The binding assay was performed by loading SA probes with either biotinylated DLK EGF1-3, biotinylated DLK EGF4-6, or biotinylated CD112 protein (negative control). An HBS buffer consisting of 0.1% BSA, 0.005% P20, was used for the experiments. The binding shift (nm) was detected by incubating the probes coated with protein (DLK1 EGF1-3, DLK1 EGF4-6 or CD112) in wells with a fixed concentration of ACVR2B (5 µM), with buffer alone as reference. The data was plotted in Prism 10 (Version 10.4.0).

### Reporting summary
Further information on research design is available in the Nature Portfolio Reporting Summary linked to this article.

## Data availability
X-ray crystallography data generated for the structure of DLK1 in complex with ACVR2B has been deposited in the Protein Data Bank under the accession code 9D20. In addition, the following crystallographic data were used in this study: 5UK5 (NOTCH1-JAG1), 4XBM (DLL1), 6MAC (ACVR2B-GDF11-ALK5), 5NH3 (ACVR2A), 1S4Y (Activin-ACVR2B), 2H64 (BMP2-ACVR2B), 5JI1 (GDF8), and 7MRZ (GDF11-ACVR2B). Unique reagents used in this study are available from the corresponding author on request. Source data are provided with this paper.

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

## Acknowledgements

We thank the staff at the 22-ID beamline of the Advanced Photon Source for assistance with remote X-ray data collection. We thank Joseph Johnson at the Analytic Microscopy Core for assistance with confocal imaging. This work has been supported in part by the Analytic Microscopy Core Facility at the H. Lee Moffitt Cancer Center & Research Institute, an NCI designated Comprehensive Cancer Center (P30-CA076292). We thank Daniel Lester and Eric Lau for assistance and access to their phase contrast microscope, Chandramohan Kattamuri for sending reagents, David Gonzalez-Perez for providing DLL4 protein, Srishti Singh for providing CD112 control protein and Charlotte Mason for managing reagents. We thank Chris Siebel for insightful advice. This work was supported by NIH R35GM133482 (V.C.L.), NIH R35GM134923 (T.B.T) and the Sigrid Juselius Foundation (D.A.). V.C.L. is a Rita Allen Scholar. Shared resources were provided by the Moffitt Cancer Center Support Grant NIH P30CA076292.

## Author contributions

D.A., V.C.L. and Q.M. wrote the manuscript. D.A., Q.M. and V.C.L. designed the experiments. D.A. performed cellular assays. Q.M., A.M and D.A. performed cloning. Q.M., A.M and D.A. produced and purified recombinant proteins. Q.M. crystallized the proteins, collected the x-ray diffraction data and performed data processing, structure solution, and refinement. Q.M., V.C.L. and D.A. analyzed the structural data. Q.M. performed SPR experiments. E.J.B. and T.B.T. provided reagents and assisted with the experimental design of TGF-β signaling assays. V.C.L. obtained funding and supervised the study. All authors read and commented on the manuscript before submission.

## Competing interests

V.C.L. is a consultant on unrelated projects for Cellestia Biotech, Remunix, and Curie. Bio. T.B.T. is a consultant/advisor for Keros Therapeutics and Oviva Therapeutics. The remaining authors have no competing interests.
