## [Transparent Peer Review file · Nature Communications]

Molecular mechanism of Activin receptor inhibition by DLK1

Corresponding Author: Dr Vincent Luca

Version 0:

Reviewer comments:

Reviewer #1

(Remarks to the Author)

In this paper, the authors demonstrate that Delta-like non-canonical Notch ligand 1 (DLK1) interacts with the TGF- β superfamily member ACVR2B, competing with TGF- β ligands for binding, thus inhibiting ACVR2B signaling. Using functional assays, the authors show that DLK1 antagonizes Myostatin-ACVR2B signaling to promote myoblast differentiation.

Having a structural resemblance to canonical NOTCH ligands, DLK1 is, as its name implies, considered a non-canonical NOTCH ligand, thought to play a modulatory role on NOTCH signaling. Despite substantial evidence, the implication of DLK1 in NOTCH signaling is controversial and other interaction partners have been suggested as well. A direct interaction between DLK1 and ACVR2B has not previously been demonstrated, although evidence exists for implication of DLK1 in TGF- β 1 signaling (Taipaleenmäki H, et al. *Stem Cells*. 2012 Feb;30(2):304-13.).

The findings of the article are highly interesting and important to the field of DLK1 biology. Although it is well established that DLK1 plays a role in developmental processes, the underlying mechanism of Dlk1 action remains to be fully elucidated.

My main concern is that in the discussion section, the authors discuss their results in relation to the role of DLK1 in adipogenesis, line 229-238. Herein, the authors state that DLK1 promotes the differentiation of adipocytes and they claim that this agrees with their model of DLK1 being an inhibitor of ACVR2B. Although some conflicting results can be found in the literature, general consensus is that DLK1 has an inhibitory effect on both proliferation and differentiation (reviewed in Traustadóttir GÁ, et al. *Cytokine Growth Factor Rev*. 2019 Apr;46:17-27 and Grassi ES, et al. *J Histochem Cytochem*. 2022 Jan;70(1):17-28.). Thus, DLK1 is widely known to negatively regulate adipocyte differentiation, instead of promoting it, therefore maintaining the preadipocyte stage. The authors need to revisit this paragraph and explain their results in the context of DLK1 being an inhibitor and not a promotor of adipogenesis.

Moreover, I find that the authors need to address the role of various DLK1 isoforms and discuss their findings in relation to the distinctive role of these isoforms especially on myoblast differentiation (Shin S, et al. *FEBS Lett*. 2014 Apr 2;588(7):1100-8.).

Minor points:

-Using SPR assay, the authors demonstrate lack of interaction between DLK1 and NOTCH receptors. In the introduction section of the manuscript, the authors challenge the relevance of two-hybrid studies for studying the interaction between DLK1 and NOTCH1 due to the lack of formation of disulfide bonds in the reducing environment of the cytosol. Can the authors elaborate on whether SPR provides a biologically relevant assay readout?

-In line 211-212 the authors write; "...may have biased the focus of DLK1 research towards Notch-related mechanisms, even in cases where Notch signaling cannot explain the results". Can the authors specify these cases?

Reviewer #2

(Remarks to the Author)

In this study the authors explored the role of the non-canonical, membrane bound ligand DLK1 in modulating Notch and TGF- β superfamily signaling. Despite prior evidence associating DLK1 with Notch signaling, through structural and

biochemical analyses, the authors demonstrated that DLK1 does not bind or activate Notch receptors directly, but instead it blocks the ligand-binding site of the TGF- β family type II receptor, ACVR2B, effectively antagonizing myostatin signaling and reversing the inhibitory effect of myostatin on myogenic differentiation. At the same time the authors provide the explanation of previous observation that DLK1 affects Notch signaling by demonstrating the presence of crosstalk involving shared downstream effectors. These findings clarify DLK1 role as a selective ACVR2B binder and antagonist, providing insights into its potential therapeutic potential in muscle growth regulation and muscle-wasting diseases.

This is a well-constructed study that provides substantial new structural and biological insights into the roles of DLK1. The crystallographic data provided appears robust and well within acceptable limits for publication. The SPR, immunofluorescence, and cell-based assay data are convincing and provide strong experimental support for the conclusions. The figures are clear and effectively highlight the key findings of the study. Overall, the quality of the manuscript and the novelty of the findings make it well-suited for publication in Nature Communications though several points need to be addressed to enhance the rigor and clarity of the work.

1. Line 143, the authors should also compare the binding constants (KD) between ACVR2B and DLK1 with those of other TGF- β family ligands, rather than focusing solely on the extent of the binding interface. This comparison would be particularly relevant given their suggestion regarding discrimination against low-affinity binders (line 225).
2. Figures 2E and S2F: The authors used yeast display to surface-express 12 different TGF- β family receptors. However, the extracellular domains of these TGF- β family receptors are rich in disulfide bonds, which could impact their proper folding on yeast. What was the rationale for choosing yeast display over U2OS or similar cells, which were used in Figure 2D?
3. The authors report that DLK1 binds to ACVR2B but not ACVR2A, highlighting amino acid differences at critical positions T93 and E94. It is important to discuss whether these differences might also impact the preference of binding of other TGF- β family ligands to these receptors. Do these mutations abolish also binding of the ligands from TGF- β family?
4. Inhibition of myostatin by DLK1 occurs primarily at high, non-physiological concentrations. Could the membrane localization of DLK1 enhance its potency? This should be investigated.
5. The authors provide nice functional data in the C2C12 cells. It would be good to knockdown (siRNA) ACVR2B, which should abolish the effect of DLK1 on the myostatin induction. For clarity in the presented data, it is essential to clearly visualize the nuclei (currently barely visible, please consider changing color) and myotubes. The authors should also include panels from day 0 and day 3 to allow for clear observation of individual myoblasts following myostatin treatment and to distinctly show the formation of myotubes. MHC staining is good but not sufficient.
6. Figure 5E middle bottom panel seems to show PLA signal inside and outside cells. Why? Please include DAPI or a similar stain to visualize the nuclei as in Supp Figure 5. As a control the authors need to show data with both antibodies alone with the PLA probes. Consider also presenting the levels of pSMAD2/3 as a readout for myostatin signaling.
7. The authors should briefly discuss what is the functional relevance of the NICD-Smad2/3 interaction. Is it necessary for the inhibition of myotube formation by myostatin?

Few technical points:

8. In Figure 1D, 4th column, the label for DLK1 should include the term 'soluble'.
9. The type of protein used should be specified more clearly throughout the text. For instance, in line 115, ACVR2B should be labeled as 'recombinant ACVR2B-Fc' to avoid ambiguity.
10. Figure S2 panels D-E, 3S panel A, S4 panels A-D – it would improve clarity to include the ligand information directly on the charts.
11. Figure 5A - should be 'ACVR1B' instead of 'ACVR1', the same correction should be done in the legend
12. Line 201 SFig. 6C should be SFig. 5C.

Reviewer #3

(Remarks to the Author)

The authors of Antfolk et al. demonstrate that DLK1, previously considered a non-canonical Notch ligand, does not physically interact with Notch, but instead binds to an Activin receptor, ACVR2B. The manuscript is relatively short and yet conceptually very straightforward, and accomplishes to convince that DLK1's previously assumed identity as a Notch ligand was likely in error. Furthermore, Antfolk et al. report the first structure of DLK bound to an activin receptor, establish binding specificity to only one Activin receptor paralog, while explaining structurally the reasons for lack of cross-reactivity. Last but not importantly, the authors show direct signaling outcomes of this new ligand-receptor interaction in *in vitro* assays, which may also provide an answer to why DLK1 was associated with Notch signaling in prior studies.

To my knowledge, these are novel results, and the manuscript addresses a major conundrum in the field. Technically, the work was expertly done. I have no major comments, and I commend the authors for providing a significant amount of supplemental data (including raw data), coordinates and map coefficients.

Below are nitpicky, minor comments that should be taken as suggestions, to help the authors:

1. Line 83: "three bulky hydrophobic interface residues (Y255, H268 and W280 in JAG1), are substituted for small hydrophobic residues (P47, S60 and G72) in DLK1"

Except for Pro, it would be a stretch to call these amino acids "hydrophobic". Instead, consider "various smaller amino acids

(P47, S60 and G72)" or something the authors would prefer.

2. Calcium has been observed to benefit Notch-Delta interactions. There is no mention of them in this manuscript, and the SPR assay buffer does not seem to include calcium ions (as previously done with some other Notch-receptor experiments). Could the authors clarify (in their response, not necessarily the manuscript) that DLK1 does not bind calcium or that calcium would not help form a putative Notch-DLK1 complex?

3. Sensorgrams for the first DLK1-ACVR2B binding experiment appears to be not cited in text and the main figure (Fig 2C), though they are present later in SFig 4A. I agree that delaying the presentation of the sensorgrams to compare with mutants is reasonable. But pointing to this supplemental panel, which is the raw data for an earlier figure, would really help readers judge this important piece of data. Maybe cite SFig 4A in the figure legend for Fig. 2C?

4. On line 130, it is mentioned, "We mapped the ACVR2B-binding domains of DLK1 using SPR, which revealed that the EGF5-6 region binds to ACVR2B with comparable affinity..."

This implies that more was done than just testing EGF5-6. While the argument is a reasonable one, it could be beneficial to include some of the negative data whether domains of DLK1 were shown to not interact with ACVR2B. The authors can decide. Also, it is not impossible that domains contributing very little to binding energy may have direct contacts (binding energy is often loaded on a few "hot spots" and not the entire interface). This relates to the only weakness of the manuscript, which is a lack of discussion for the entire complex beyond the domains characterized, and the oligomeric state and architecture of the complex.

5. On line 156, the authors mention, "This F82 residue is 156 substituted for a smaller isoleucine (I83) residue in ACVR2A, likely disrupting hydrophobic packing." While this is a very reasonable interpretation, would the authors also consider the Phe->Ile change causing a clash as the beta-branched isoleucine clashes with the tightly packed loop in DLK1 around residue Ala 198? This can be seen by mutating Phe to Ile in PyMOL or any other visualization software. This is entirely for the authors to decide.

6. I found the relationship between Fig. 5B and Suppl. Fig. 5A a bit confusing. It may help to explain this a bit more in the main text, especially for the uninitiated. A more descriptive y-axis in Fig. 5B could help.

7. I might have missed this, but I could not find the sensorgrams for the SPR runs in SFig. 5B. These should be added to the supplement.

8. On line 187, did the authors mean Fig. 5B instead of 5C?

9. I assume by HBS-P20-BAS on line 382, the authors meant HBS-P20-BSA. Regardless, though, this acronym does not appear to be used elsewhere, so it may be removed if it is not ever invoked?

10. Based on the coordinates, maps and statistics provided by the authors, the structure is high quality, and was competently determined. One small point: Flipping the carbonyl oxygen in chain D residue 197 (as in chain B) would remove a Ramachandran outlier and get rid of unwanted difference density.

11. The authors can fix small stylistic issues across the manuscript, such as inconsistent use of uM instead of μ M, missing subscript in "MgCl₂" or the use of the German letter eszett (β) instead of beta, β , in TGF- β .

Version 1:

Reviewer comments:

Reviewer #1

(Remarks to the Author)

The authors have adequately addressed all my comments in the revised version of the manuscript. Therefore, I have no further comments.

Reviewer #2

(Remarks to the Author)

I am pleased with the revisions made in response to the initial comments. The authors have clearly made a substantial effort to address all concerns in a thorough and thoughtful manner. The additional data, particularly the experiments involving membrane-bound DLK1 and additional controls (Bimagrumab), have strengthened the conclusions and provided a clearer understanding of the biological relevance of their findings. Furthermore, the improved imaging and inclusion of new supplementary figures have greatly enhanced the clarity of the manuscript. Additionally, substantial efforts have been done to refine the discussion.

The manuscript is now well-organized, clearly written, and provides robust new insights into DLK1 biology. The only minor oversight is that in Figure 5A the labelling still reads "ACVR1" instead of "ACVR1B," This should be corrected. Overall, I consider the manuscript significantly improved and now suitable for publication.

Reviewer #3

(Remarks to the Author)

The authors have addressed all of my comments satisfactorily. I especially appreciate adding the new panel Suppl. Fig. 3A. This manuscript strongly demonstrates that the proposed Notch ligand DLK1 is in fact an Activin receptor ligand instead. I have no remaining major or minor concerns about the work. The study's results are significant and important for the field.

RESPONSE TO REVIEWERS' COMMENTS

We thank all the reviewers for their comments about the importance of our study to the field and for taking time to assess our manuscript. We have revised the manuscript to address nearly all reviewer concerns, and we believe it has improved substantially due to the helpful feedback.

Reviewer #1 (Remarks to the Author):

In this paper, the authors demonstrate that Delta-like non-canonical Notch ligand 1 (DLK1) interacts with the TGF- β superfamily member ACVR2B, competing with TGF- β ligands for binding, thus inhibiting ACVR2B signaling. Using functional assays, the authors show that DLK1 antagonizes Myostatin-ACVR2B signaling to promote myoblast differentiation.

Having a structural resemblance to canonical NOTCH ligands, DLK1 is, as its name implies, considered a non-canonical NOTCH ligand, thought to play a modulatory role on NOTCH signaling. Despite substantial evidence, the implication of DLK1 in NOTCH signaling is controversial and other interaction partners have been suggested as well. A direct interaction between DLK1 and ACVR2B has not previously been demonstrated, although evidence exists for implication of DLK1 in TGF- β 1 signaling (Taipaleenmäki H, et al. Stem Cells. 2012 Feb;30(2):304-13.).

The findings of the article are highly interesting and important to the field of DLK1 biology. Although it is well established that DLK1 plays a role in developmental processes, the underlying mechanism of Dlk1 action remains to be fully elucidated.

My main concern is that in the discussion section, the authors discuss their results in relation to the role of DLK1 in adipogenesis, line 229-238. Herein, the authors state that DLK1 promotes the differentiation of adipocytes and they claim that this agrees with their model of DLK1 being an inhibitor of ACVR2B. Although some conflicting results can be found in the literature, general consensus is that DLK1 has an inhibitory effect on both proliferation and differentiation (reviewed in Traustadóttir GÁ, et al. Cytokine Growth Factor Rev. 2019 Apr;46:17-27 and Grassi ES, et al. J Histochem Cytochem. 2022 Jan;70(1):17-28.). Thus, DLK1 is widely known to negatively regulate adipocyte differentiation, instead of promoting it, therefore maintaining the preadipocyte stage. The authors need to revisit this paragraph and explain their results in the context of DLK1 being an inhibitor and not a promotor of adipogenesis.

In this paragraph, our goal was mainly to emphasize that the DLK1-ACVR2B interaction may play a role in processes beyond myogenesis. However, we did not provide much detail about the logic behind these statements, and we agree that it was confusing. We updated the paragraph to more clearly reflect our thought process (relevant sentences below):

“DLK1 has mostly been shown to inhibit adipogenesis, although a subset of studies suggest that DLK1 may promote adipogenesis in certain contexts. It will be interesting to investigate the functional relevance of DLK1 as a TGF- β -family ligand inhibitor in these and other developmental systems.” & “We speculate that DLK1 may have differential effects depending on the presence of high- or low-affinity ACVR2B ligands based on its moderate affinity for ACVR2B. However, this will need to be tested experimentally in future studies.”

Moreover, I find that the authors need to address the role of various DLK1 isoforms and discuss their findings in relation to the distinctive role of these isoforms especially on myoblast differentiation (Shin S, et al. FEBS Lett. 2014 Apr 2;588(7):1100-8.).

We agree that this is important to address, and we have updated the discussion to mention a possible explanation for our findings in light of the study by Shin et al. The new sentences read,

"In a previous study, a soluble isoform of mouse DLK1 was an ineffective inhibitor of myoblast differentiation compared to a slight promoter by membrane-bound isoforms compared to untreated cells. By contrast, our data show that DLK1 inhibits the negative effects of myostatin both in its cellular and soluble forms. In addition to the fundamental difference in experimental setup, we attribute this difference to the high concentrations of recombinant DLK1 required for inhibition in our assays, as compared to the presumably lower concentrations secreted from transfected cells, although it's also possible that the multiple DLK1 isoforms found in mice have functions beyond the two isoforms found in humans."

At the suggestion of reviewer 2, we have also added new experimental data, where we express DLK1 on the membrane of our CAGA-SMAD reporter cells. We found that transfection of DLK1 inhibited myostatin-induced SMAD signaling in the reporter line (Fig. 5C, Suppl. Fig. 5D & 9A-C) In light of this new data, the discussion expanded naturally to include the above comparison of soluble and membrane-tethered DLK1.

Minor points:

-Using SPR assay, the authors demonstrate lack of interaction between DLK1 and NOTCH receptors. In the introduction section of the manuscript, the authors challenge the relevance of two-hybrid studies for studying the interaction between DLK1 and NOTCH1 due to the lack of formation of disulfide bonds in the reducing environment of the cytosol. Can the authors elaborate on whether SPR provides a biologically relevant assay readout?

SPR is not necessarily more biologically relevant, rather, the proteins used in our SPR assay were purified in eukaryotic systems that enable them to fold properly and form biologically relevant interactions. In our study, the microscopy-based staining assays and flow data provide complementary validation of the SPR and show DLK1 binds to ACVR2B in a more native context. Together with our inability to observe any SPR signal for DLK1 binding to NOTCH1, or DLK1 staining of NOTCH1/2/3 overexpressing cells, this strongly implies the lack of a meaningful extracellular interactions between DLK1 and Notch-receptors.

-In line 211-212 the authors write; "...may have biased the focus of DLK1 research towards Notch-related mechanisms, even in cases where Notch signaling cannot explain the results". Can the authors specify these cases?

Although we intended the sentence as a general statement, where the consensus mechanism of DLK1 as a Notch regulator may deter researchers from exploring other pathways, we provide a few examples (see below) where this is evident:

1) Grassi et al., studied hypoxia-mediated effects of DLK1 and tried to link the effects through Notch by testing reporter assays and looking for the effect through RNA data. However, the authors end up noting:

“Our findings join a third category of reports suggesting that DLK1 effects are independent on Notch: DLK1-overexpressing cells were no different in their activation of Notch signaling compared with controls in a reporter assay, and DLK1 expression in no way correlated with expression of classical Notch downstream target genes in human GBM, as analyzed in TCGA data. In spite of the close structural relation between DLK1 and canonical Notch ligands, mechanisms of DLK1 signaling in glioma appear to be more complex than inhibition of Notch receptor activation.”

2A)

Additionally, most investigations do not detail unsuccessful binding attempts or other negative data. However, for DLK1 and Notch some negative data exists, and we can only speculate how many have tried and failed to report similar data before the first case was published. Wang et al., tested IP of DLK1 (Pref-1) and Notch and show lack of any interaction as noted in their Figure 1, titled “Pref-1 does not interact with Notch or require Notch for its signaling.”

2B)

Despite the description of DLK1 as ligand for Notch1, in their paper “Evidence of non-canonical NOTCH signaling: Delta-like 1 homolog (DLK1) directly interacts with the NOTCH1 receptor in mammals” Traustadottir et al., also disclose “However, in agreement with previous reports [32], [46], we failed to co-immunoprecipitate NOTCH1 and DLK1 despite using three different approaches (data not shown). This may be explained by DLK1 function depending on unknown co-factors or physical properties of the DLK1-NOTCH1 interaction not present in every setting.”

We simply think that our alternative mechanism should be considered instead of focusing on differences in physical properties or unknown factors. We have added Grassi et al., and Wang et al., as references to this sentence in the discussion.

Reviewer #2 (Remarks to the Author):

In this study the authors explored the role of the non-canonical, membrane bound ligand DLK1 in modulating Notch and TGF- β superfamily signaling. Despite prior evidence associating DLK1 with Notch signaling, through structural and biochemical analyses, the authors demonstrated that DLK1 does not bind or activate Notch receptors directly, but instead it blocks the ligand-binding site of the TGF- β family type II receptor, ACVR2B, effectively antagonizing myostatin signaling and reversing the inhibitory effect of myostatin on myogenic differentiation. At the same time the authors provide the explanation of previous observation that DLK1 affects Notch signaling by demonstrating the presence of crosstalk involving shared downstream effectors. These findings clarify DLK1 role as a selective ACVR2B binder and antagonist, providing insights into its potential therapeutic potential in muscle growth regulation and muscle-wasting diseases.

This is a well-constructed study that provides substantial new structural and biological insights into the roles of DLK1. The crystallographic data provided appears robust and well within acceptable limits for publication. The SPR, immunofluorescence, and cell-based assay data are convincing and provide strong experimental support for the conclusions. The figures are clear and effectively highlight the key findings of the study. Overall, the quality of the manuscript and the novelty of the findings make it well-suited for publication in Nature Communications though several points need to be addressed to enhance the rigor and clarity of the work.

1. Line 143, the authors should also compare the binding constants (KD) between ACVR2B and DLK1 with those of other TGF- β family ligands, rather than focusing solely on the extent of the binding interface. This comparison would be particularly relevant given their suggestion regarding discrimination against low-affinity binders (line 225).

We agree that the surface area bound is not necessarily indicative of inhibition potency and we have changed the text in this discussion. It is somewhat difficult to compare exact affinities of canonical TGF-beta ligands due to differences in experimental conditions, assay design, the use of Fc-dimers, kinetic vs steady-state analysis etc. Given that we see functional inhibition of Myostatin, which is a high affinity ligand to ACVR2B, we expect that ligands with lower reported affinities than myostatin would be more effectively blocked regardless of the exact reported affinities. In one direct comparison between Myostatin, BMP2 and BMP7, the BMP-ligands' had up to 88-fold lower affinity for ACVR2B compared to Myostatin (BMP2 is ~88-fold lower, BMP7 is ~4-fold lower) (Sako et al., JBC, 2010).

At the suggestion of the reviewer, we also tested the ability of membrane DLK1 to inhibit myostatin-ACVR2B. This data was added to the main text as Fig. 5C and shows that cells transfected with DLK1 have reduced SMAD reporter activity in response to myostatin stimulation. This suggests that the high local concentration of DLK1 restricted to the membrane may also compensate for its relatively lower affinity.

2. Figures 2E and S2F: The authors used yeast display to surface-express 12 different TGF- β family receptors. However, the extracellular domains of these TGF- β family receptors are rich in disulfide bonds, which could impact their proper folding on yeast. What was the rationale for choosing yeast display over U2OS or similar cells, which were used in Figure 2D?

Although U2OS cells would have also been appropriate, we used yeast because of the high-throughput nature of the system, and because eukaryotic yeast cells are capable of forming disulfide bonds/glycosylating extracellular proteins. For this reason, yeast display is preferable to phage display for complex extracellular proteins. Several studies have now used yeast display for a variety of extracellular, disulfide-containing proteins (see Fig. 1 of Gai & Wittrup. 2008, Curr Opin Struct Biol), and our own group has published several studies where we successfully displayed disulfide-rich Notch ligands and immune checkpoint proteins on yeast (Gonzalez-Perez et al. 2023, Nat Chem Biol, Ming et al. 2022, Nat Immunol).

3. The authors report that DLK1 binds to ACVR2B but not ACVR2A, highlighting amino acid differences at critical positions T93 and E94. It is important to discuss whether these differences might also impact the preference of binding of other TGF- β ligands to these receptors. Do these mutations abolish also binding of the ligands from TGF- β family?

We are not aware of examples where any canonical ligands are deficient in ACVR2A binding to the level of DLK1. However, we found that sequence alignment and structural analysis provides insight into the importance of these residues for biased recognition. In ACVR2A, E94 is substituted with K95 at the analogous interface position (see below). This charge reversal is not expected to substantially affect binding to the uncharged serine or threonine residues in Activin A or B. On the other hand, the interface residues predicted to be opposite E94 in Activin C/E are negatively charged glutamates, suggesting that these ligands would prefer ACVR2A. Lastly, these same

interface residues are lysines in GDF8 and GDF11, suggesting that they prefer ACVR2B based on charge complementarity with E94.

Our analyses are consistent with reports showing that GDF11 and GDF8 preferentially bind ACVR2B over ACVR2A (Lee & McPherron, 2001, PNAS; Goebel et al., 2019, PNAS), and that Activin C and Activin E preferentially bind ACVR2A (Goebel et al., 2022, eLife). We have updated the manuscript with new supplementary data figure panels (Fig. S4 H-I). The relevant panels are also shown below.

Reviewer figure 1. Structural analysis of ACVR2A and ACVR2B ligand-binding residues. Structures of GDF11 bound to ACVR2B (PDB: 7MRZ) and an AlphaFold model of Activin C bound to ACVR2A show the predicted positions of interface residues. The sequence alignment on the right highlights the residues predicted to contact E94 in ACVR2B or K95 in ACVR2A.

4. *Inhibition of myostatin by DLK1 occurs primarily at high, non-physiological concentrations. Could the membrane localization of DLK1 enhance its potency? This should be investigated.*

We agree about the informative value of such an experiment, and we decided to examine the effect of DLK1 at the surface of cells by transfection of full length DLK1. We used our CAGA-reporter cell line, previously used to inhibit myostatin signaling by soluble DLK1 to give a comparison between soluble DLK1 and DLK1 at the membrane.

We find that full length DLK1 transfected into the CAGA-reporter cell line, with 90% of the cells expressing DLK1 at the surface, led to a reduction in myostatin mediated SMAD-signaling by approximately 50% and thus corresponds roughly to the IC50 value of around a 2.8 μ M concentration of soluble DLK1 in the same readout. Thus, DLK1 expressed at the surface can influence myostatin signaling to levels achieved with what can be considered a high concentration of soluble ligand.

We have added this new inhibition data as a main result in Figure 5C and a repeat with additional controls in Suppl. Fig. 5D. The expression levels of DLK1 at the surface after transfection was added to Suppl. Fig. 9A-C.

5. The authors provide nice functional data in the C2C12 cells. It would be good to knockdown (siRNA) ACVR2B, which should abolish the effect of DLK1 on the myostatin induction. For clarity in the presented data, it is essential to clearly visualize the nuclei (currently barely visible, please consider changing color) and myotubes. The authors should also include panels from day 0 and

day 3 to allow for clear observation of individual myoblasts following myostatin treatment and to distinctly show the formation of myotubes. MHC staining is good but not sufficient.

Regarding ACVR2B knockdown: this would likely abolish the effect myostatin has on myotube formation altogether, and would thus preclude rescue with DLK1. However, we believe we were able to address the reviewer's concern using a different control by testing the effect of the ACVR2B antagonist antibody Bimagrumab (BYM338). We found that adding Bimagrumab (at a lower dose than DLK1) had the same effect as DLK1 on differentiation, and that adding DLK1 + Bimagrumab did not lead to further differentiation, as might be the case if they affected differentiation through distinct mechanisms. We would also like to point out that our null-ACVR2B binding DLK1 R193D mutant was ineffective in the differentiation assay, strongly supporting our interpretation of the results. For the new supplementary images of Bimagrumab as a control compared to soluble DLK1, we added the Day 0 and Day 3 images as brightfield images to complement fluorescent images as the reviewer had suggested (Suppl. Fig. 7).

Additionally, we have improved the clarity of the differentiation images by pseudo colored nuclei and 'zoom in' panels of myotubes. We have further included longer time frame (96h) differentiation images in the supplement (Suppl. Fig. 6B) to help the reader and reviewer examine the myotube formation better.

We have also taken utmost care to avoid repeated culturing and introduction of foreign material through transfection in the C2C12 cells, as multiple passages of C2C12s would be required to confidently establish a modified C2C12 line. C2C12 cells are known to lose their differentiation capability over time and are very sensitive to cell-cell contact through confluency. For all assays in this manuscript, we have purchased C2C12 cells directly from ATCC and used them only between passages p3-p7 (considering ATCC cells p1) and cultured them under 10% confluency until the set-up of an experiment.

With the improvement and increased clarity of the main figure resulting from better visualization of the nuclei and myotubes, we would like to keep the main figure in place. We think the low magnification of the wells that show large amounts of myotubes is a transparent way of visualizing these results. Due to low contrast in most brightfield images taken with the same field of view and objective as our fluorescent images, we did not image brightfield for all samples, and our previous optimization of the assay showed strong overlap between MyoHC staining and brightfield views. However, we decided to add a new supplementary figure showing representative images of the optimization of the imaging at different time-points of differentiation.

We found that our MyoHC staining and unstained myotubes correlate strongly, but with lower contrast in visualized brightfield images. In fact, we observed that in certain cases — particularly when myotubes are either very large or absent — they are readily detectable by MyoHC staining but can be difficult or nearly impossible to discern using brightfield imaging (This can be exemplified in our new Suppl. Fig 6B, in the middle panel of the bottom row, showing myotubes stained at 96 hours of differentiation, where the largest myotube cluster on the right is apparent in MyoHC-green staining, but almost indistinguishable in the brightfield view, especially without the visual aid of the fluorescent staining (see Suppl. Fig. 6B).

We hope that the additional data clarify this point and adequately address the reviewer's concern.

6. Figure 5E middle bottom panel seems to show PLA signal inside and outside cells. Why? Please include DAPI or a similar stain to visualize the nuclei as in Supp Figure 5. As a control the authors need to show data with both antibodies alone with the PLA probes. Consider also presenting the levels of pSMAD2/3 as a readout for myostatin signaling.

There are no PLA signals outside any cell, but instead the actin staining used to show the contours of that one cell in the middle panel has slightly weaker fluorescence compared to others. The cell that the reviewer is referring to is also slightly behind the other cells spatially. A very faint green cytoskeleton can still be seen in the original figure around the PLA dots. Using the reviewer's suggestion of providing images of nuclei staining in the main figure makes it clear that these PLA signals are all inside the cell. However, we provide an image for the reviewer only with a significantly increased green actin signal to demonstrate also the presence of the cell skeleton around the PLA signals. When the balance of the whole image is considered, we think that this wildly oversaturates all the other cells and prefer to keep a level of fluorescence in the figure where the majority of cells have a normal level of actin fluorescence that won't overpower the image or detract from the main result of showing the PLA signals. These updated images are now in Figure 6B. We have included our negative controls for both antibodies alone in the supplementary file as requested. These are now in Suppl. Fig. 8A-B.

Reviewer figure 2. Increased intensity of green fluorescence representing the actin cytoskeleton show actin contour around PLA dots.

While we understand the usefulness of pSMAD levels in relation to these assays, we put extensive efforts into optimizing our SMAD assay using the CAGA-reporter cell line, which include full dose-response curves with DLK1 concentrations up to 16 μ M. Together with the new data showing inhibition mediated by transfected DLK1 in these same cells (now included in the main figure), we hope the reviewer finds the signaling data satisfactory.

7. The authors should briefly discuss what is the functional relevance of the NICD-Smad2/3 interaction. Is it necessary for the inhibition of myotube formation by myostatin?

This is good point as it could be implied that this signaling is key for the myotube formation, which is not exactly our intent. We think that the effect of myostatin on muscle development is reasonably well defined and depends on SMAD signaling through regulation of myogenic factors such as MyoD, MYOG and Atrogin-1 along with additional effects through indirect signaling such as MAPK-related downstream effectors.

We mainly wanted to see if this crosstalk also happens in the C2C12 myoblasts, similarly to what has already been shown by immunoprecipitation in HEK293 cells (*Blokzijl et al., 2003, J Cell Biol.*). With that said we think it's certainly a possibility that the NICD - SMAD interaction could synergistically regulate MyoD and MYOG expression in light of studies such as (Kuroda et al., 1999, J Biol Chem) and have added discussion about this to the manuscript.

1. Lee SJ, McPherron AC. (2001). "Regulation of myostatin activity and muscle growth." *Proc Natl Acad Sci USA.* 98(16): 9306–9311.
2. Tintignac et al. (2005). "Degradation of MyoD mediated by the SCF (MAFbx) ubiquitin ligase." *J Biol Chem.* 2005 Jan 28;280(4):2847-56.
3. Elkina, Y., von Haehling, S., Anker, S.D. et al. The role of myostatin in muscle wasting: an overview. *J Cachexia Sarcopenia Muscle* 2, 143–151 (2011). <https://doi.org/10.1007/s13539-011-0035-5>.

Few technical points:

8. In Figure 1D, 4th column, the label for DLK1 should include the term 'soluble'.

We have updated this description to improve clarity.

9. The type of protein used should be specified more clearly throughout the text. For instance, in line 115, ACVR2B should be labeled as 'recombinant ACVR2B-Fc' to avoid ambiguity.

We have updated the text in multiple instances, highlighted in the text.

10. Figure S2 panels D-E, 3S panel A, S4 panels A-D – it would improve clarity to include the ligand information directly on the charts.

We have added this information.

11. Figure 5A - should be 'ACVR1B' instead of 'ACVR1', the same correction should be done in the legend

We have updated the figure and legend.

12. Line 201 SFig. 6C should be SFig. 5C.

We have corrected this error.

Reviewer #3 (Remarks to the Author):

The authors of Antfolk et al. demonstrate that DLK1, previously considered a non-canonical Notch ligand, does not physically interact with Notch, but instead binds to an Activin receptor, ACVR2B. The manuscript is relatively short and yet conceptually very straightforward, and accomplishes to convince that DLK1's previously assumed identity as a Notch ligand was likely in error. Furthermore, Antfolk et al. report the first structure of DLK bound to an activin receptor, establish binding specificity to only one Activin receptor paralog, while explaining structurally the reasons for lack of cross-reactivity. Last but importantly, the authors show direct signaling outcomes of this

new ligand-receptor interaction in in vitro assays, which may also provide an answer to why DLK1 was associated with Notch signaling in prior studies.

To my knowledge, these are novel results, and the manuscript addresses a major conundrum in the field. Technically, the work was expertly done. I have no major comments, and I commend the authors for providing a significant amount of supplemental data (including raw data), coordinates and map coefficients.

Below are nitpicky, minor comments that should be taken as suggestions, to help the authors:

1. Line 83: "three bulky hydrophobic interface residues (Y255, H268 and W280 in JAG1), are substituted for small hydrophobic residues (P47, S60 and G72) in DLK1"

Except for Pro, it would be a stretch to call these amino acids "hydrophobic". Instead, consider "various smaller amino acids (P47, S60 and G72)" or something the authors would prefer.

We agree - we have updated the text in line with this comment.

2. Calcium has been observed to benefit Notch-Delta interactions. There is no mention of them in this manuscript, and the SPR assay buffer does not seem to include calcium ions (as previously done with some other Notch-receptor experiments). Could the authors clarify (in their response, not necessarily the manuscript) that DLK1 does not bind calcium or that calcium would not help form a putative Notch-DLK1 complex?

Although DLK1 does not have calcium-binding EGF repeats annotated in uniprot, we did use CaCl_2 in our NOTCH1 experiments. We thank the reviewer for noticing and apologize for the omission of our buffer composition for the Notch SPRs, as we had only included the buffer for our ACVR2B to DLK1 experiments. For NOTCH1 experiments, we used a buffer including 1 mM CaCl_2 and have updated the materials and methods to reflect this omission. We have also added details for producing the Notch proteins, which were also supplemented with CaCl_2 during production and purification.

3. Sensorgrams for the first DLK1-ACVR2B binding experiment appears to be not cited in text and the main figure (Fig 2C), though they are present later in SFig 4A. I agree that delaying the presentation of the sensorgrams to compare with mutants is reasonable. But pointing to this supplemental panel, which is the raw data for an earlier figure, would really help readers judge this important piece of data. Maybe cite SFig 4A in the figure legend for Fig. 2C?

We have taken this suggestion and now refer to Suppl. Fig. 4A in the figure legend so that readers who want to inspect the sensograms can find this underlying data more easily (as long as the journal allows for this representation). We appreciate that the reviewer noticed that by delaying the sensogram, we can later effectively compare it directly to the sensograms of the interface mutations.

4. On line 130, it is mentioned, "We mapped the ACVR2B-binding domains of DLK1 using SPR, which revealed that the EGF5-6 region binds to ACVR2B with comparable affinity..."

This implies that more was done than just testing EGF5-6. While the argument is a reasonable one, it could be beneficial to include some of the negative data whether domains of DLK1 were

shown to not interact with ACVR2B. The authors can decide. Also, it is not impossible that domains contributing very little to binding energy may have direct contacts (binding energy is often loaded on a few "hot spots" and not the entire interface). This relates to the only weakness of the manuscript, which is a lack of discussion for the entire complex beyond the domains characterized, and the oligomeric state and architecture of the complex.

We agree with the reviewer that showing negative data can be beneficial for the domain mapping discussion. In our initial pilot studies, we measured binding using a single concentration of ACVR2B to EGF1-3 or EGF4-6 of DLK1 using biolayer interferometry (BLI) and found that only EGF4-6 binds. We then used SPR to measure the affinity between EGF5-6 and the full ECD of DLK1 and found that it was nearly identical to the affinity between full-length DLK1 and EGF5-6, so we ultimately included that data in the manuscript. We have now updated the manuscript to include the BLI data (both positive and negative). The new sentence reads:

We mapped the ACVR2B-binding domains of DLK1 using biolayer interferometry (BLI). We determined that ACVR2B interacts with DLK1 EGF4-6, but not with EGF1-3 (Suppl. Fig. 3A), and SPR measurements further revealed that the EGF5-6 region binds to ACVR2B with comparable affinity ($K_D = 1.0 \mu\text{M}$) to the full-length ECD (Suppl. Fig. 3B).

Regarding the full-length complex, we agree that this needed to be discussed in more detail. We added the following sentence to the end of the 2nd paragraph of the discussion to address the possibility that other aspects of DLK1 are functionally important. "Although we show that DLK1 and ACVR2B can interact in the absence of additional components, we cannot exclude the possibility of modulation by co-regulatory proteins or higher-order complex formation in a cellular environment. We also speculate that the intracellular domain of DLK1 may contribute additional regulatory functions. Future studies will be necessary to fully understand the full architecture and regulatory mechanisms of this complex."

5. On line 156, the authors mention, "This F82 residue is substituted for a smaller isoleucine (I83) residue in ACVR2A, likely disrupting hydrophobic packing." While this is a very reasonable interpretation, would the authors also consider the Phe->Ile change causing a clash as the beta-branched isoleucine clashes with the tightly packed loop in DLK1 around residue Ala 198? This can be seen by mutating Phe to Ile in PyMOL or any other visualization software. This is entirely for the authors to decide.

We agree with the observation that Ile 83 could also clash with Ala 198 and have added this comment in the manuscript. In figure 4B, we highlighted Ala 198 in addition to Val 209 and Val 229.

6. I found the relationship between Fig. 5B and Suppl. Fig. 5A a bit confusing. It may help to explain this a bit more in the main text, especially for the uninitiated. A more descriptive y-axis in Fig. 5B could help.

We have updated these figures with more descriptive information. In the main figure we have a fitted dose response curve, whereas we show the raw luciferase assay values as bar graphs in the supplementary, with the bars for maximum myostatin activity and maximum inhibition next to each other. The main figure is based on the left graph in the supplementary, but we wanted to show the values of all titrations independently as bar graphs.

7. I might have missed this, but I could not find the sensorgrams for the SPR runs in SFig. 5B. These should be added to the supplement.

We have added the sensogram, it is now accompanying Suppl. Fig. 5B as 5C.

8. On line 187, did the authors mean Fig. 5B instead of 5C?

Yes, we have updated the manuscript accordingly.

9. I assume by HBS-P20-BAS on line 382, the authors meant HBS-P20-BSA. Regardless, though, this acronym does not appear to be used elsewhere, so it may be removed if it is not ever invoked?

We have fixed the error with BSA.

10. Based on the coordinates, maps and statistics provided by the authors, the structure is high quality, and was competently determined. One small point: Flipping the carbonyl oxygen in chain D residue 197 (as in chain B) would remove a Ramachandran outlier and get rid of unwanted difference density.

We flipped the carbonyl oxygen of residue 197 in chain D, and this removed a Ramachandran outlier as the reviewer had astutely observed. We have updated the final pdb-file with this correction for publication. We appreciate the thorough examination of our structure.

11. The authors can fix small stylistic issues across the manuscript, such as inconsistent use of uM instead of μ M, missing subscript in "MgCl2" or the use of the German letter eszett (β) instead of beta, β , in TGF- β .

We have replaced all β to β , uM to μ M, um to μ m, ug to μ g and MgCl₂ with the number in subscript.

REVIEWERS' COMMENTS

Reviewer #1 (Remarks to the Author):

The authors have adequately addressed all my comments in the revised version of the manuscript. Therefore, I have no further comments.

Reviewer #2 (Remarks to the Author):

I am pleased with the revisions made in response to the initial comments. The authors have clearly made a substantial effort to address all concerns in a thorough and thoughtful manner. The additional data, particularly the experiments involving membrane-bound DLK1 and additional controls (Bimagrumab), have strengthened the conclusions and provided a clearer understanding of the biological relevance of their findings. Furthermore, the improved imaging and inclusion of new supplementary figures have greatly enhanced the clarity of the manuscript. Additionally, substantial efforts have been done to refine the discussion.

The manuscript is now well-organized, clearly written, and provides robust new insights into DLK1 biology. The only minor oversight is that in Figure 5A the labelling still reads "ACVR1" instead of "ACVR1B," This should be corrected. Overall, I consider the manuscript significantly improved and now suitable for publication.

We have updated the figure and the legend to say ACVR1B.

Reviewer #3 (Remarks to the Author):

The authors have addressed all of my comments satisfactorily. I especially appreciate adding the new panel Suppl. Fig. 3A. This manuscript strongly demonstrates that the proposed Notch ligand DLK1 is in fact an Activin receptor ligand instead. I have no remaining major or minor concerns about the work. The study's results are significant and important for the field.